

# Near-surface characterization and delineation of water conduits at South Deep Gold Mine, South Africa

Sikelela Gomo[1], Farbod Khosro Anjom[2], Chiara Colombero[2], Mohammadkarim Karimpour[2], Bibi Ayesha Jogee[1], Musa S.D. Manzi[1], Laura V. Socco[2,3]

[1]School of Geosciences, University of the Witwatersrand, Johannesburg, Private Bag, Wits, 2050, Republic of South Africa.

[2]Department of Environment, Land and Infrastructure Engineering, Politecnico di Torino, Turin, 10129, Italy.

[3]Department of Geoscience and Engineering, Delft University of Technology, Delft.

*Correspondence to*: Sikelela Gomo (1488846@students.wits.ac.za)

**Abstract.** Velocity models of the shallow (a few hundred of meters) geology are hardly retrieved from deep seismic reflection exploration data, despite their importance in near-surface characterization, improving seismic mapping resolution at depth, and constraining deeper geological models. In this work, we compute the near-surface shear wave velocity model in the vicinity of South Deep Gold Mine, using surface waves present in the 2D and 3D deep seismic reflection data acquired at the mine. The obtained near-surface model is then used to (1) characterize the near-surface

and (2) better constrain possible water conduits (faults, fracture zones and dykes) mapped at mining levels, that enable the migration of water from overlying formations to the mining levels, and (3) constrain the timing of faults and dykes activity in the vicinity of the mine. The analysis is carried-out on reflection seismic data acquired for deep mineral exploration, where the acquisition parameters were not optimized for surface wave techniques, thus the reciprocity principle is used to improve the data density, coverage, and near-surface mapping resolution. The lithostructural

information retrieved from the produced pseudo-2D and -3D shear wave velocity models are consistent with information obtained from available borehole data and published records in the study area. Integrating the produced near-surface shear wave velocity model with a legacy 2003 P-wave seismic reflection cube, mine mapping and drilling information enabled the investigation of the structural linkage between the near-surface groundwater aquifers and deep mining levels (~ 3 km depth). The faults and dykes mapped at the mining level intersect and cross the near-

surface aquifers, thus making these structures possible conduits for water migration to the deep mining levels. The findings of this research illustrate the advantages of integrating shallow and deep subsurface information to constrain the timing of geological structures and mitigate the risks associated with water ingress to the mining levels. The final model produced can be used for future mine development, improving safety and production, and for extending the Life of Mine (LoM).



## 1 Introduction

The Mesoarchean-aged Witwatersrand Basin hosts the largest known gold deposit in the world and has produced more gold than any other ore province globally (Frimmel, 2019). Seismic techniques have been used extensively in the exploration of the Witwatersrand Basin, mainly for mapping and evaluating gold-bearing horizons (locally termed 'reefs') for mining purposes (i.e., planning, production, and risks mitigation) and for imaging faults and dykes that might (1) act as possible conduits for water and methane migration into the mining levels, and (2) offset the orebody (Campbell and Crotty, 1988; Pretorius et al., 1989; Gibson, 2005; Manzi et al., 2012). The first 3D application of seismic techniques in hard-rock mining environments, particularly for mineral exploration, trace back to the surveys conducted by Campbell and Cotty (1988, 1990) at South Deep Gold Mine in 1986 in the Witwatersrand Basin. Campbell and Crotty (1990) applied active-source surface 3D seismic survey to image the gold-bearing horizons and geological structures at depths down to 3.5 km below ground surface. This pilot study investigated and demonstrated the potential that 3D seismic technique has in hard-rock environment for mineral exploration. Between the years 2022 and 2023, detailed 2D and 3D deep seismic reflection (DSR) surveys, i.e., exploration of the Witwatersrand Basin, were conducted at South Deep Gold Mine, located in the West Rand Goldfield (South Africa), as part of the ERA-MIN3 Future project (Rapetsoa et al., 2025). The work mainly focused on imaging the deep (~ 3.5 km) gold-bearing horizons such as the Ventersdorp Contact Reef (VCR), Upper Elsburg Reefs (UER), and complex geological structures (e.g., faults and dykes) that cross-cut the orebodies. The seismic surveys used a seismic nodal system incorporating one vertical and three component recorders connected to 5 Hz geophones, broadband micro-electromechanical system (MEMS) accelerometers, distributed acoustic sensing technology, and a 6-ton broadband (2-200 Hz) seismic vibrator operating with sweep lengths between 24s and 48s. In this work, we utilize the surface waves (SWs) present in the DSR survey data acquired at the mine to estimate the near-surface shear wave (S-wave) velocity model at the vicinity of the mine, which is then use to (1) better understand the near-surface stratigraphic (lithological) and structural (e.g., faults and dykes) geological variations in the study area, and (2) investigate the structural linkage between shallow geology (aquifers) and deep mining levels where the orebody is located to constrain the timing of activity for structures and determine potential conduits for water ingress into the mining levels.

The near-surface is a complex environment, often consisting of heterogenous structures characterized by highly weathered and diverse materials, i.e., that may vary from loose, fractured, to solid rocks, which have the potential to significantly obscure the ability to resolve deeper lying seismic targets (Sheriff, 2002; Zhou et al., 2010; Strong, 2018). Near-surface characterization is an essential process in exploration, mining, engineering, and environmental studies as it can provide information that can be used to improve the imaging of deeper lying targets and structures, and conducting risk assessments, efficient project planning, and for designing and constructing safer and better-performing mining, environmental, and civil engineering infrastructure (Lakke et al., 2008; Zhu et al., 2008; Socco et al., 2010; Pegah and Liu 2016; Papadopoulou et al., 2020). The application of SW analysis to determine near-surface velocity models at hard-rock sites is limited due to the challenge of obtaining good-quality data (Socco et al., 2019; Papadopoulou et al., 2020). The quality of SWs at hard-rock mining sites can be severely affected by (1) the presence of lateral heterogeneities (e.g., dykes, fractures, and faults) which can generate scattering and back reflections (Pileggi et al., 2011; Malehmir, 2015); (2) noise generated by mining activity (e.g., blasting, excavations, drilling, crushing,




moving heavy machinery, and mining operations) (Urosevic et al., 2007; Górszcyk et al., 2015); and (3) variable and extreme near-surface conditions caused by the presence of hard and rugged topography, roads, areas of outcropping crystalline rock, and the presence of swampy soils or dense vegetation (Saunders et al., 1991; Eaton et al., 2003; Heinonen et al., 2011;  Malehmir et al., 2017). However, the dominance of SWs in seismic records; their high sensitivity to near-surface properties; their ability to resolve low-velocity zones or soft layers bounded by high-velocity materials; and their ability to avoid the water-masking effect in saturated media, make them a powerful tool for near-

surface imaging (Foti et al., 2002; Socco and Strobbia, 2004).

This study presents the application of SW analysis to reflection seismic data mainly acquired for deep targeting in a hard-rock mining environment. Multichannel analysis of surface wave (MASW) and the Laterally Constrained Inversion (LCI) are applied to high-resolution 2D and 3D seismic data to estimate the near-surface S-wave velocity model within the top ~ 360 m in the vicinity of South Deep Gold Mine. Since the deep exploration data acquired at

the mine is not optimized for SW analysis (i.e., characterized by coarse spatial sampling) and is therefore not ideal for SW analysis, we utilize the reciprocity theorem to extract all the possible information in the data, i.e., increase the data coverage by providing a grid of orthogonal sets of dispersion curves along receiver and source (shot) lines. Comparison with available borehole data in the vicinity of the mine indicates that the conducted analysis is sensitive to stratigraphic and structural variations beneath the acquired DSR surveys and resolves them quite well with depth.

Lastly, we integrate the obtained near-surface S-wave velocity model with existing and available legacy 2003 active-source P-wave seismic reflection data, mine mapping and drilling information to investigate the structural continuity and connectivity between the shallow aquifers (< 500 m) and deep mining levels (~3-3.5 km). These mine-mapped structures are well known at South Deep Gold Mine to be associated with mining-induced seismicity, water migration, and methane gas pockets. The integration of the near-surface S-wave model with the deeper datasets enabled these

structures to be tracked (vertically) through the thick 3 km strata from their known positions at the mining levels to the near-surface.

In this paper, we (1) describe the mining site and its geological setting, (2) present the seismic data acquisition parameters and the characteristics of the data, (3) outline the processing and inversion strategy, and (4) discuss the obtained velocity models in connection with borehole data, mine-mapped structures and legacy deep seismic data.


### 2 Site and data

South Deep Gold Mine is situated southwest of the city of Johannesburg, in the southern portion of the West Rand Goldfield and on the northern margin of the Witwatersrand Basin (Fig. 1a and b). Excluding the Archaean basement rocks of the Kaapvaal Craton and volcanic and sedimentary rocks of the Dominion Group underlying the

Witwatersrand Supergroup, present in the West Rand Goldfield are the rocks of the, from oldest to youngest, Witwatersrand, Ventersdorp, and Transvaal Supergroups (Kositcin and Krapež, 2004). The rocks of the Ventersdorp Supergroup are unconformably overlain by the Vryburg Formation/Black Reef Quartzite of the Chuniespoort Group, the basal Group of the late Archean to early Proterozoic Transvaal Supergroup (Martin et al., 1998). The Transvaal



Supergroup is a thick succession of chemical and sedimentary rocks, hosting well-preserved volcanic and glaciogenic
units (Bekker et al., 2014; Erikson et al., 1993) and world-class iron (Fe) and manganese (Mn) ores (Bekker et al.,
2014; Smith and Beukes, 2016; Franchi, 2018) and comprises of the lower Chuniespoort and upper Pretoria Groups
(Fig. 1b; Martin et al., 1998). The Chuniespoort Group dolomites consist of Karst systems rich in water, which is
primarily responsible for water inflow and inrush into mine shafts and mining levels (Van Niekerk and Van Der Walt,
2006).

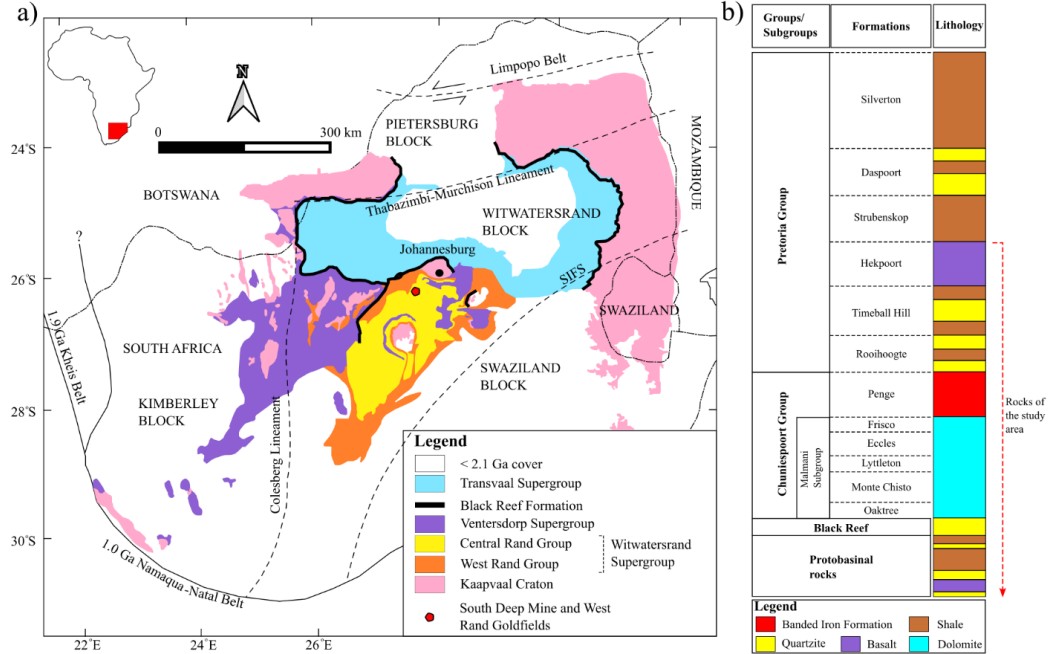

**Figure 1: (a) Simplified geological map showing both surface and subsurface distribution of the main Archaean stratigraphic units of the Kaapvaal Craton. The map shows the distributions of the Witwatersrand (comprised of the West Rand and Central Rand Groups), Ventersdorp and the Transvaal Supergroups (modified after Frimmel, 2014). (b) Generalized stratigraphic column of the uppermost rocks in the Transvaal Supergroup, indicating the uppermost rock units outcropping in the study area.**

Predominantly outcropping in the vicinity of South Deep Mine are the rocks of the early Proterozoic Pretoria Group
(2.35-2.25 Ga; Walraven et al., 1990; Martin et al., 1998). The Pretoria Group forms the upper part of the Transvaal
Supergroup and primarily consists of a succession of shales, quartzose and arkosic sandstones, and significantly
interbedded basaltic-andesitic volcanic rocks and subordinate conglomerates, diamictites and carbonate rocks (Killick,
1992; McCarthy, 2006). The sedimentary units of the Pretoria Group were deposited by a combination of coastal
processes, rivers, alluvial fans, fan-deltas, and basinal turbiditic and suspension sedimentation systems (Eriksson et
al., 1993a, b, 1995a). All rocks of the Pretoria Group have been subjected to low-grade metamorphism and a general



sheet-like geometry is evident for most of the formations, with certain sandstones and lava units exhibiting wedge-like three-dimensional forms (Eriksson et al., 2006). In the northern portion of the South Deep Mine, rocks of the Malmani Subgroup, Timeball Hill Formation, and Klapperkop Quartizites are present, while south of the mine, outcrops of basaltic lava and tuff of the Hekpoort Formation exist (Fig. 2a and b; Osburn et al., 2014). In Figure 2c, we show the borehole logs drilled in proximity to the 2D and 3D seismic data acquired in the study area between 2022 and 2023 (see Figures 2a and 2b for borehole location). The logs include the lithological variation within the top 360 m of the study area and are later used to constrain the conducted SW analysis interpretation. The borehole logs illustrate that, south of the mine, the ~2230 $\pm$ 13 Ma (Burger and Coertze, 1973) Hekpoort Formation basalts constitute the uppermost rocks and are underlain by the 2324$\pm$17 Ma (Dorland et al., 2004) metasedimentary rocks of the Timeball Hill Formation. The rocks of the Pretoria Group are displaced by post-Pretoria Group faults that displace both the Pretoria Group rocks and the base of the Transvaal Supergroup, and slump-faulting, which only affects the formations within the Transvaal Supergroup (Cousins, 1962). The slump faulting does not transgress beyond the base of the Transvaal Supergroup and is thought to originate from the subsidence of the Chuniespoort Group of the Transvaal Supergroup. Additionally, the Transvaal Supergroup is displaced by post-Transvaal Supergroup age faults associated with the Vredefort Impact Crater (Cousins, 1962) and is intruded by massive suites of mafic and ultramafic rocks (i.e., dyke swarms and sill provinces) thought to be 'pre'- 'syn'- and 'post'-Bushveld Complex in age (Willemse, 1959; Sharpe, 1982, 1984; Schreiber et al., 1992; Gumsley et al., 2017). These post-Transvaal Supergroup age faults and intrusions are prone to mining-induced seismicity and influence the hydrology of the West Rand acting as pathways for water migration from the overlying Chuniespoort Group dolomitic units down to the gold reefs' mining levels (~3.5 km below ground surface) (Van Niekerk and Van Der Walt, 2006; Manzi et al., 2012). The intersection of these water conduits during mining often negatively affects the productivity of the mine and increases the safety risk to mine personnel and infrastructure. Thus, their delineation is important in ensuring mine safety, longevity, and increased productivity.





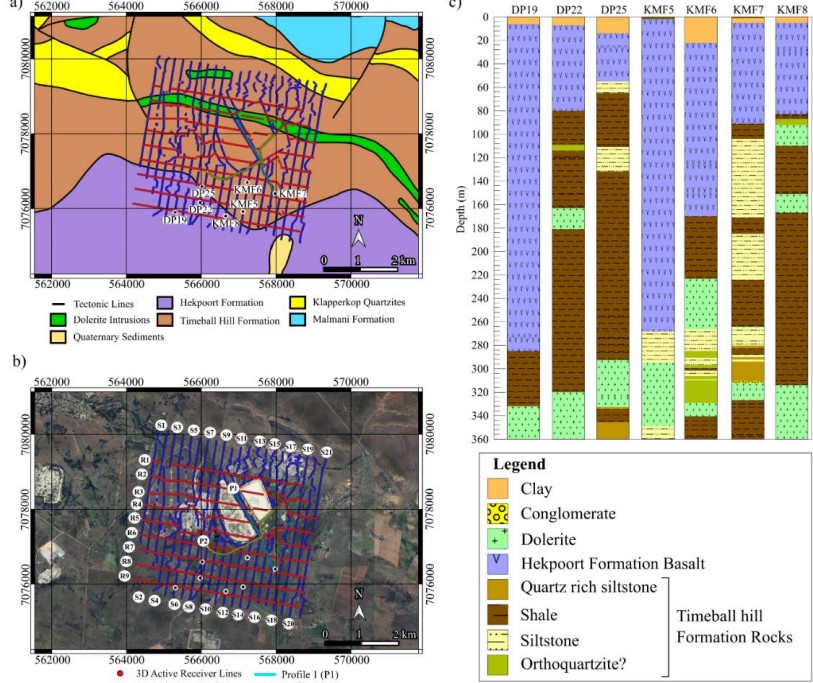

**Figure 2: (a) High-resolution 2D profiles (P1-blue and P2-yellow), and the receiver (red) and source (dark blue) lines of the seismic 3D grid acquired at the mine overlain on the local geological map of the study area. (b) Satellite map showing South Deep Gold Mine and seismic data geometry. Both the geological and satellite maps, i.e., (a) and (b), are plotted using the WGS 1984 UTM Zone 35S coordinate system. (c) Representative borehole logs located within the study area and showing the dominant rock types of the top 360 m. Borehole locations are reported in Figure 2a and b.**

## 3 Seismic data acquisition

Active-source surface 2D and 3D seismic data were acquired within the framework of the ERA-MIN3 Future Project, aimed at imaging the subsurface geology and the orebody at South Deep Gold Mine (Fig. 1). Two 2D high resolution seismic lines (P1 and P2, Fig. 2a and b) were positioned such that they traversed the surface zone directly above the mine tunnels, located at a depth of ~ 3.5 km. The 3D seismic survey was designed to cover a wider area of approximately 4.4 km by 4.7 km, with the mine tunnels located at the centre of the grid. The acquisition parameters are summarized in Table 1. The 2D profiles were acquired using a 6-ton seismic vibrator (mini-vibe) employing a linear sweep of 24 s and a sweep frequency ranging from 2-200 Hz. P1 is north-south orientated and has a profile length of 2420 m, while P2 is east-west orientated and has a profile length of 2650 m. The two profiles were acquired using 243 and 266 wireless remote acquisition units spaced 10 m apart and connected to 5 Hz vertical component geophones, respectively. Two sweeps were generated at every receiver and every second receiver location for P1 and P2, respectively.





The 3D seismic survey was acquired using two broadband mini-vibes employing a sleep-sweep acquisition technique with a sleep time of 30 s, single linear sweep of 48 s, and frequency ranging from 4 to 150 Hz. The grid consisted of 9 receiver lines (north-south orientated and 400 m apart), with a receiver spacing of 25 m, and 21 source lines (east-west orientated and 225 m apart), and with a source spacing increment of 12.5 m, forming an orthogonal acquisition patch. The data were recorded using a total of 1605 wireless remote acquisition units connected to 5 Hz vertical component geophones.

The units used to record both the 2D and 3D surveys were programmed to switch on at 8:00 am and off at 17:00 pm, South African time. During the acquisition period, i.e., 8:00 am to 17:00 pm, the seismic data were recorded in continuous mode, at a 2 ms sampling rate. After the completion of the survey acquisition stage, all the separate sweeps were later isolated and 6 s of the records were extracted from the recording units using the GPS time tagged for each sweep. The layouts of the surveys, however, were distorted by natural (river and swamps) and manmade features (mining infrastructure, two large tailings dams, roads, and nearby settlements) present in the vicinity of the mine, thus, sharp turns and gaps were introduced in the data (Fig. 2b). The study area is characterized by a significant but gentle E-W elevation variation of approximately 100 m in the southern portion, a gentle $\pm$ 10 m E-W elevation variation in the northern section, and a $\pm$ 20 m N-S elevation variation in the middle section of the study area. Throughout the study area, the bedrock is largely covered by residual soil.

**Table 1**: Acquisition parameters of the 2D and 3D surveys conducted at South Deep Gold Mine

| Survey Parameters | Profile 1 (P1) | Profile 2 (P2) | 3D Grid |
|---|---|---|---|
| Profile length | 2420 m | 2650 m | 4.5 x 4.7 km |
| Type | - | - | Sleep-sweep, sleep time: 30 s |
| Sampling rate | 2 ms | 2 ms | 2 ms |
| Sweep length | 24 s | 24 s | 48 s |
| Sweep frequencies | 2-200 Hz | 2-200 Hz | Linear, 4-150 Hz, custom sweep |
| Receiver Direction | N-S | E-W | E-W |
| Shot Direction | N-S | E-W | N-S |
| Shot spacing | 10 m | 20 m | 12.5 m |
| Receiver spacing | 10 m | 10 m | 25 |
| Shot Stack | 2 | 2 | - |
| Receiver line spacing | - | - | 400 m |
| Shot line spacing | - | - | 225 m |
| Geophone | 5 Hz vertical component | 5 Hz vertical component | 5 Hz vertical component |
| Source type | Mini-vibe (6-ton) | Mini-vibe (6-ton) | Mini-vibe (6-ton) |
| Geodetic survey instrument | Differential Global Positions System (DGPS) | Differential Global Positions System (DGPS) | Differential Global Positions System (DGPS) |

185



## 4 Data Processing

The data were subdivided into three datasets (Fig. 2b; Table 1): the two 2D lines P1 and P2; nine E-W oriented receiver lines (R1-R9) from the 3D grid; and twenty-one S-N oriented source (shot) lines (S1-S21) from the 3D grid (Fig. 2). The source line dataset was generated from the receiver line dataset by using the reciprocity principle to generate common receiver gathers (CRG) along the source lines. The reciprocity principle states that a wavefield measured at point A generated by a point source located at point B is equal to the wavefield measured at point B generated by a point source located at point A (Knopoff and Gangi, 1959; Claerbout, 2008; Katou et al., 2017). In seismic, it illustrates that the same seismogram will be recorded if seismic source and receiver locations are interchanged, irrespective of the geometrical subsurface complexity, as long as the geometry of obstacles and other bodies in the vicinity of the source and receiver are fixed (Claerbout, 2008). The implementation of the reciprocity theorem was achieved by selecting receivers situated along and near source lines and using them to generate CRG by projecting waveforms recorded at the receivers to their respective shot location.

MASW was applied to the three seismic datasets described above using a workflow based on the method by Socco et al. (2009), where a moving spatial window is used along the seismic lines to retrieve a set of local SW dispersion curves (DCs). Figure 3 gives a schematic summary of the entire processing workflow. The raw data (1), undergo a data preconditioning (2) that consists in splitting the profiles characterized by severe sharp turns into separate segments close to straight lines, as severe profile crookedness distorts the phase shift variation between neighboring traces and results in spurious velocities (Lin et al., 2017). The selection of optimal MASW processing parameters (3), i.e., spatial window length, overlap of the moving spatial window, the maximum offset range for shot selection were defined to obtain a good compromise between spectral and spatial resolution and the optimal signal-to-noise ratio for the dispersion curve analysis by doing some test on the different datasets. (Socco et al., 2009). The spatial window needs to be long enough to ensure the recording of longer wavelengths of interest (i.e., the target depth) and small enough to not violate the homogenous media assumption assumed underneath the receiver spread. Meanwhile, the minimum and maximum offset ranges are chosen such that they minimize near-and far-field effects and improve the signal-to-noise ratio. For each window position, the spectra resulting from different shots falling within the selected offset ranges were computed using the phase shift method (Park et al., 1998) and stacked to improve the signal-to-noise ratio of the dispersion image (4) (Grandjean and Bitri, 2006; Neducza, 2007). From the stacked spectra, the maximum spectral amplitude at each frequency, deemed to correspond with the energy of the fundamental mode, was picked to obtain the local DCs (5). The DCs were picked manually to ensure high-quality DC picks. The positions of the DCs were assigned as the center of the windows for which the dispersion spectrum was computed.



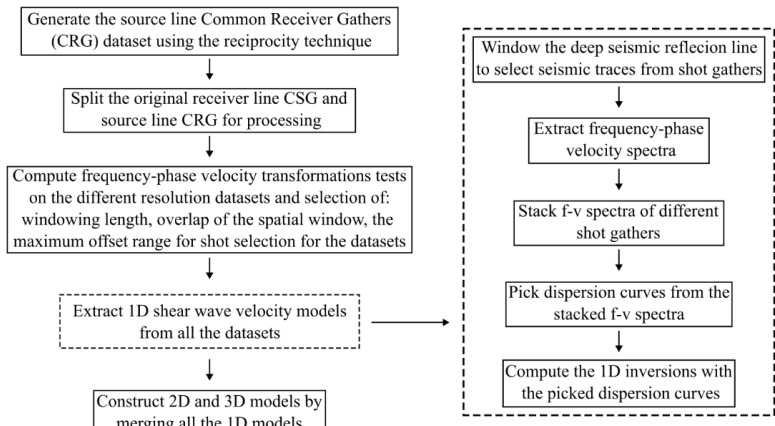

**Figure 3: Schematic summary of the entire processing procedure**.

In our case the three datasets have different spatial sampling characteristics that lead to different processing parameter choice (Table 3). For all datasets, relatively large spatial windows are selected to window the data due to the large spatial sampling used to acquire the data, which could result in more local anomalies being suppressed. The 2D lines P1 and P2 are those with the denser spatial sampling (10 m receiver spacing) and shot layout (10-20 m shot spacing) this allowed to obtain high signal-to-noise spectra and broad band DCs on 250 m windows and 200 m max offset range, while for receiver lines R1-R9 the shorter window size that provided good quality DC was 500 m. The shift of the moving window position was set equal to the receiver spacing and this led to different density of DC along the lines for the three datasets. For the two datasets extracted from the 3D grid, the most critical aspect was related to the small number of available sources in line with the receivers (or the opposite for the CRGs). The shot spacing along receiver lines is 225 m (distance between shot lines), while for the CR gathers the "shot" spacing is 400 m (distance between receiver lines) leading to very limited stacking of the spectra.

**Table 2**: Windowing parameters used for the optimal dispersion curve extraction.

| Processing parameters | 2D Line | Normal 3D grid receiver lines | On Reciprocity lines (source lines) |
|---|---|---|---|
| Minimum window size | 250 m | 500 m | 125 m |
| Maximum window size | 250 m | 500 m | 250 m |
| Window stride | 10 m | 25 m | 12.5 m |
| Minimum offset | 10 m | 10 m | 10 m |
| Maximum offset | 200 m | 400 m | 400 m |

Figure 4a shows an example of a shot gather obtained in the study area, along profile P2. The seismic reflection record in Figure 4a contains a broad band ground roll that is clearly visible at near offset traces. Figure 4b gives the frequency-phase velocity spectrum obtained after transforming the data in Figure 4a into the frequency-phase velocity domain.





The spectrum was obtained after stacking spectrum from 5 different shots within the set offset ranges, where the DC
      (fundamental mode) is picked as the spectral maxima. Figure 4c shows all the picked DCs for P1 and P2, while Figures
      4d and e display them as a function of wavelength at their spatial location along the lines. In Figures 4d and e, it is
      remarkable that long wavelengths, i.e., high investigation depths, are achieved thanks to the high velocities prevalent
      at the site. Figures 5a, b and c show the DCs that were picked along the 3D grid receiver lines, source lines, and

receiver and source lines combined, respectively.

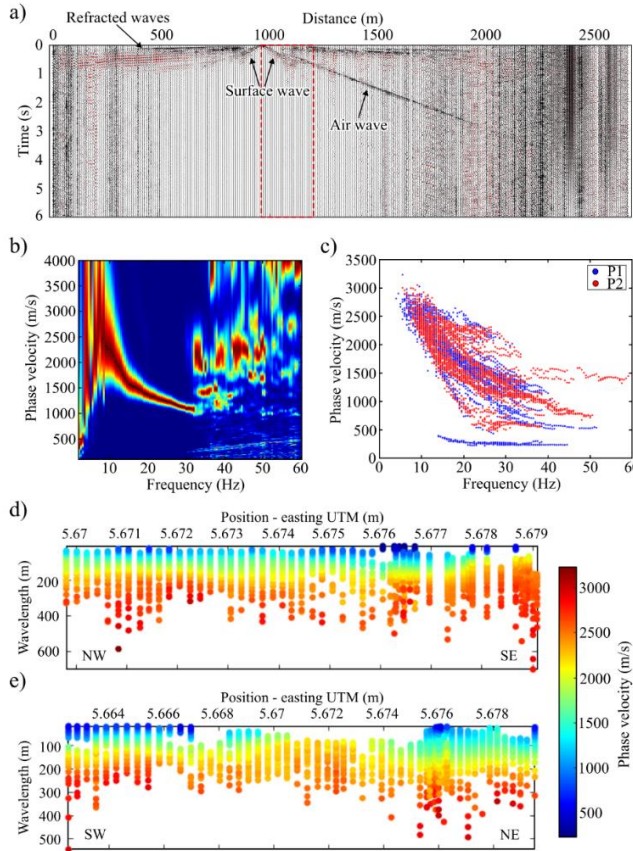

**Figure 4: (a) Shot gather from the South Deep Mine, taken from P2 and showing the window position that is later
transformed into the frequency-phase velocity domain, and (b) frequency-phase velocity spectrum from P2 obtained after
transforming the highlighted data in Figure (a), where the black dots depict the picked dispersion curve. (c) Picked surface**

**wave dispersion curves for profiles P1 (blue) and P2 (red), plotted as a function of frequency and phase-velocity. In (d) and
(e) surface wave dispersion curves for profiles P1 and P2 plotted as a function of wavelength depth and spatial location.**

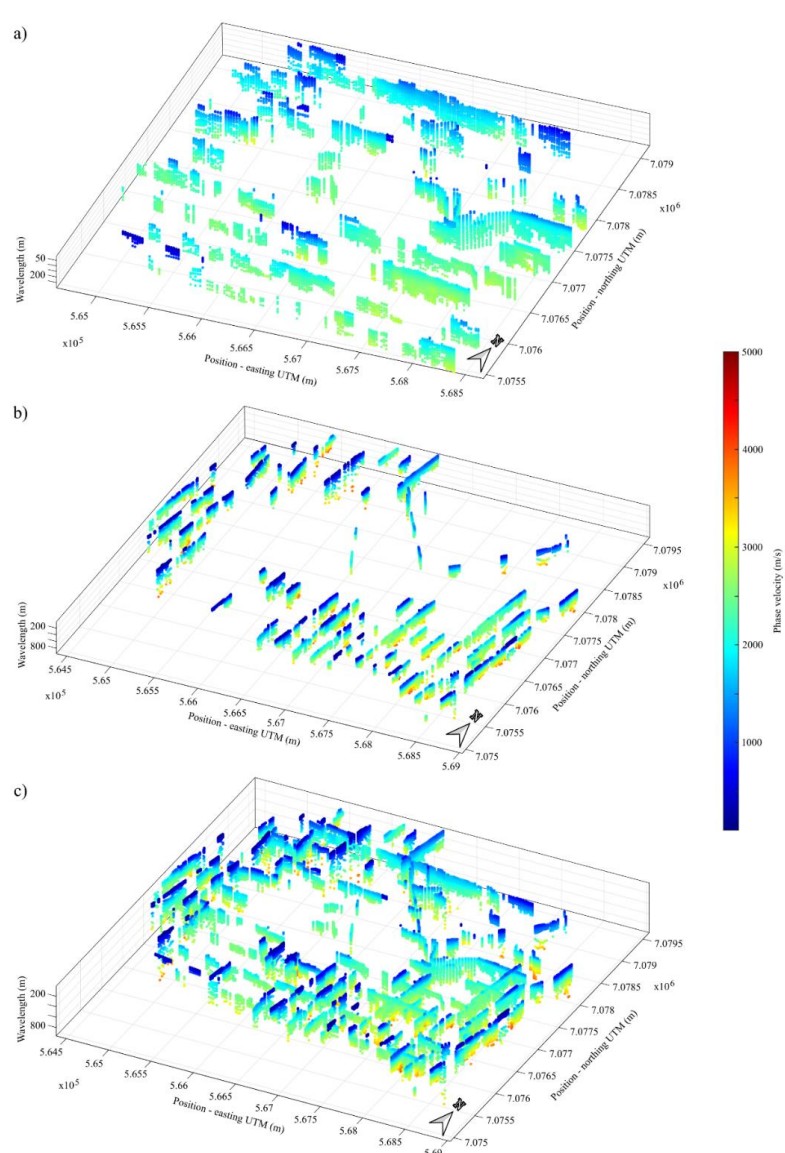

**Figure 5: Dispersion curves picked along the: (a) receiver lines; (b) source lines and (c) combined receiver and source lines.**


A total number of 2557 DCs were picked along the 3D grid, 767 of the DCs were extracted along receiver lines, while 1790 (more than twice) were extracted along source lines. The application of the reciprocity principle enabled the DC density to be increased by ~ 230 % rounding-off. In the retrieved DC datasets, gaps are present in areas: (1) where the stacked spectra did not show a clear fundamental mode, thus no DC could be picked, which is possibly due to the presence of sharp lateral variations within the subsurface, (2) where data could not be acquired due to the presence of




mining infrastructure (e.g., tailings dams, satellites, roads, etc.), and (3) where source and receiver lines are distorted by mining infrastructure.

Figure 6a gives an example of a CRG obtained for line S19. To validate the dispersion curves obtained from the CRG processing, we compared the spectra located at the crossing point for source and receiver lines (Fig. 6b). The

comparison needs to be evaluated taking into consideration that the spectra are computed along two orthogonal directions and therefore the considered wavefields sample different portions of the subsurface, that given the large window length can lead to different phase velocities. We therefore do not expect the spectra to be necessarily equal. In general, there is a relatively good agreement between the spectra of the two orthogonal lines and spectra computed from CRG are broader band than those computed along receiver lines. This is attributed to the traces in CRGs having

a smaller spacing (Table 1). Meanwhile, the spectra from the receiver lines have greater spectral resolution, due to the spectra along these lines being computed using larger spatial windows.

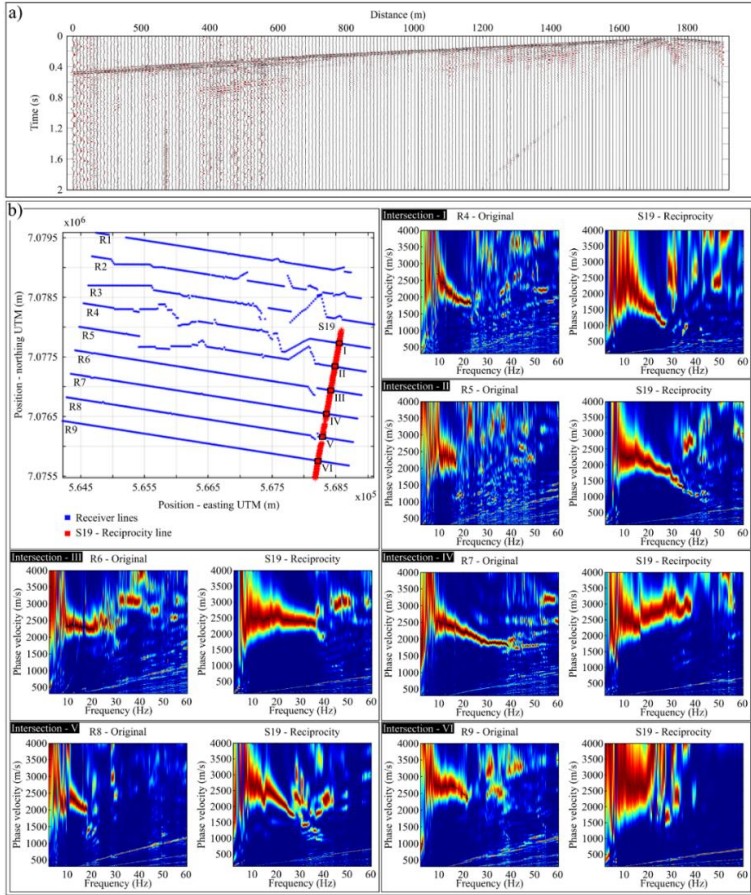

**Figure 6: (a) Example of a common receiver gather obtained along source line S19. (b) Example of receiver and source line dispersion curves obtained at intersection locations.**




In Figure 7a-f, we show the wavelength coverage of the estimated DCs for the 3D grid as pseudo-slices within different wavelength ranges, i.e. for progressively higher investigation depths (considering approximately depth = wavelength/2.5). The highest coverage is achieved for wavelengths between 50 m and 350 m, but it is still significant up to 550 m over wide areas of the grid.

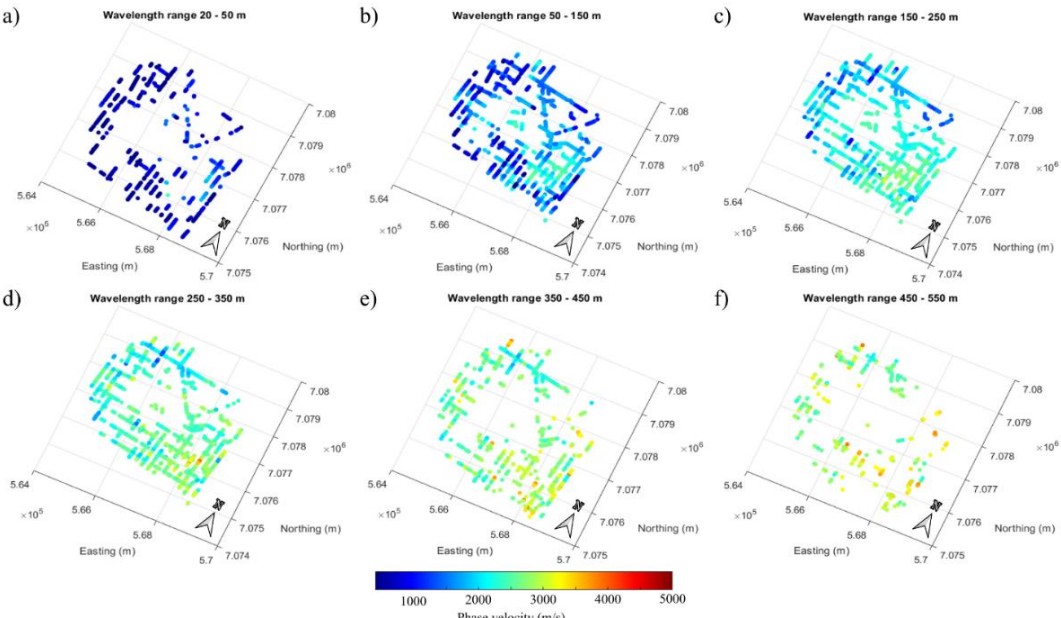


**Figure 7: Pseudo-slices of the estimated dispersion curves from the site shown using wavelength ranges of (a) 20 - 50 m, (b) 50 - 150 m, (c) 150 - 250 m, (d) 250 - 350 m, (e) 350 - 450 m, and (f) 450 - 550 m.**

## 5 Laterally constrained inversion (LCI)

To invert the picked DCs for S-wave velocities, we adopted the LCI scheme. The LCI was first proposed by Auken and Christiansen (2004) for inverting resistivity data using a pseudo-2D layered parameterization. The LCI is a deterministic inversion technique that is predicated on an algorithm created for a 1D scheme that makes use of lateral constraints. A lateral constraint is a parameter that defines the variance permitted for parameters of neighbouring models (Wisén and Christiansen, 2005; Socco et al., 2009). The constraints represent a spatial regularization that

avoids overfitting the final model. The smaller the expected variance in the area, the stronger the lateral constraints (Wisén and Christiansen, 2005; Socco et al., 2009). The optimization of the lateral constraints can be done by running a set of inversion starting from unconstrained and rising gradually the level of constraints until the fitting between model and data start to decrease (Boiero and Socco, 2010). The inputs are the DCs and the initial model at the locations of the DCs. The initial model parameters of the LCI inversion scheme are the layer thicknesses, Poisson's ratio and S-

wave velocities. The technique simultaneously inverts all the picked SW DCs by minimizing a common objective



function, which incorporates the fitting of data with model, the a priori information, and the constraints. The LCI uses a damped least-squares inversion scheme to update the model iteratively. The initial S-wave velocities and layer thickness are updated after each iteration, while layer densities and Poisson's ratio are kept fixed (Khosro Anjom et al., 2024). The final result of the LCI is a S-wave pseudo-2D/3D velocity model.

In this study, the parameterization of the initial model is laterally invariant, Table 3, and is based on the obtained SW information and expected geological properties of the site: The initial S-wave velocity model was created from the picked DCs, while Poisson's ratio and density measurements were estimated based on the materials expected in the study area (i.e., mainly the partially metamorphosed sedimentary rocks of the Timeball Hill Formation). After some inversion tests, we used a lateral constraint of 700 m/s, which was the highest level of constraint that could invert the
data without underfitting or overfitting the picked DCs. Figure 8 illustrates an example of the fitting obtained between picked and modelled DCs for lines R1, R2, and R3 (Fig. 8a), along with the global misfit calculated between the inverted and real DCs for each iteration (Fig. 8b). The lowest global misfit obtained after 39 iterations is 65 m/s.

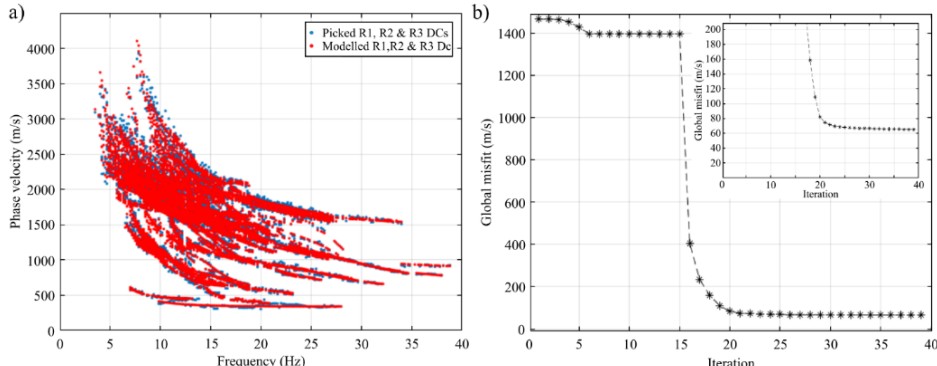

**Figure 8: (a) Fitting between picked and modelled (from last iteration) R1, R2, and R3 dispersion curves. (b) Misfit**
**between the picked and modelled dispersion curves in (a).**

Table 3: Initial model parameters used to invert the picked dispersion curves.

| Layers | S-wave velocity (m/s) | Density ($g/cm^3$) | Poisson's ratio | Thickness (m) |
|---|---|---|---|---|
| 1 | 1000 | 1.8 | 0.3 | 20 |
| 2 | 1020 | 1.8 | 0.3 | 30 |
| 3 | 1200 | 1.8 | 0.3 | 30 |
| 4 | 1500 | 1.8 | 0.3 | 30 |
| 5 | 1500 | 1.8 | 0.3 | 30 |
| 6 | 1700 | 1.8 | 0.3 | 30 |
| 7 | 2000 | 1.8 | 0.3 | 30 |
| 8 | 2500 | 1.8 | 0.3 | 50 |
| 9 | 3000 | 1.8 | 0.3 | 100 |



## 6 Results and Interpretation

Figure 9 shows the near-surface S-wave velocity field models obtained through the LCI of the picked DCs along the 2D lines (P1 and P2, Figs. 9a, b) and 3D grid, i.e., along the receiver and source lines (Fig. 9c). In Figures 9, 10 and 11, both vertical and lateral velocity variations in the study area are resolved.

### 6.1 Lithological layering, faults and intrusions

From the obtained high-resolution 2D S-wave velocity models, for P1 and P2 (Figs. 9a, b), notable is that the site is characterized by a prominent (1) velocity reversal zone – indicated with a black solid line in Figure 9b and (2) low- and high-velocity zones that extend from the surface to great depths (i.e., > 150 m). Evident in Figure 9c is that the S-wave velocity reversal zone, high and low velocity zones observed in Figures 9a and b are resolved in the 3D grid S-wave velocity model as well (Fig. 9c). A good S-wave velocity and anomaly spatial location correlation is observed

between the 2D and 3D models within the high-velocity, low-velocity and velocity reversal zones (Fig. 9c). In the 3D S-wave velocity model, the velocity reversal zone is overlain by a high-velocity zone with a sheet-like geometry (i.e., near-horizontal and planar-orientated). This sheet-like high-velocity zone is better observed in the zoom of Figure 11.





**Figure 9: Shear wave velocity models of the (a) high-resolution 2D lines (i.e., P1 and P2), (b) their 3D plot and (c) 3D grid data plot. The dashed black-grey, black-pink, and black-red circles indicate correlating high, low and velocity reversal zones between the obtained 2D and 3D shear wave velocity models, respectively.**

Figure 10 illustrates the correlation between available borehole logs in the study area and the S-wave velocity profiles closest to them. Apparent in Figure 10a-c is that the identified velocity reversal zone (indicated by a solid black line) marks the boundary/contact between the overlying basalts of the Hekpoort Formation and the underlying shales and



siltstones of the Timeball Hill Formation. Also notable in Figure 10a-d is that where the depth extent of the uppermost Hekpoort Formation rocks is greater than ~ 50 m and less than ~ 200 m, the contact between the Hekpoort Formation and Timeball Hill Formation is mapped relatively well; while in areas where the thickness of the Hekpoort Formation is greater than ~ 200 m, the Hekpoort-Timeball Hill Formation contact cannot be resolved, see Figures 10d and e.

Indicated by the black dashed lines are the significant lateral variations present within the rocks of the Hekpoort and Timeball Hill Formations, which are elaborated more on in the section that follows. This comparison of S-wave velocity profiles and proximal borehole information illustrates that the obtained S-wave velocities are sensitive to the lithological variation characteristic of the site and are reasonable and reliable.

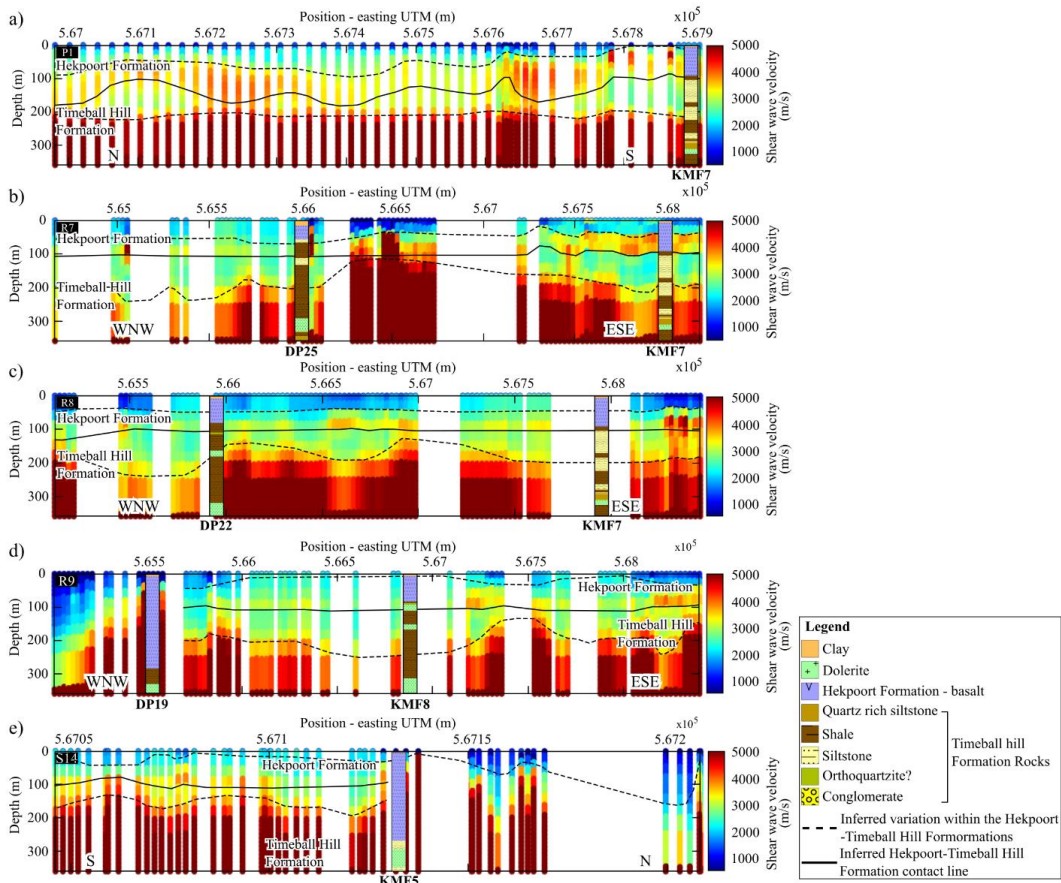

**Figure 10: Representative boreholes of the study area projected on pseudo-2D shear wave velocity models closest to them. Pseudo-2D shear wave velocity models of lines: (a) P1, (b) R7, (c) R8, (d) R9 and (e) S14.**

In general, within the top 360 m of depth, the overall S-wave velocity is interpreted to show four sheet-like subsurface layers (Figs. 10 and 11a). The first layer roughly extends from the surface to a depth of ~ 40 m and has S-wave velocity

ranging between < 500 m/s to ~ 2500 m/s. The second layer shows higher velocities (~ 2500 m/s to ~ 3500 m/s) and extends from approximately 40 m to 120 m below the surface. A third layer, with S-wave velocities similar to the first





one, extends from approximately 120 m to 200 m. The last resolvable layer is a high-velocity zone (> ~3500 m/s), extending from approximately 200 m of depth. The first two identified layers belong to the Hekpoort Formation and are interpreted to constitute the weathered and least weathered zones of the Hekpoort Formation rocks, respectively. The last two layers belong to the upper shales of the Timeball Hill Formation, the high velocities in the second layer of the Timeball Hill Formation are attributed to the presence of sills at those depths, see Figure 2. During the emplacement of ultramafic-mafic rocks of the Transvaal Supergroup, shale- and quartzite-dominated formations (i.e., Timeball Hill, Boshoek, and Daspoort Formations) were found to form preferential sill emplacement locations as compared to the carbonate and volcanic formations of the Supergroup. The shales and quartzite formations being preferential emplacement locations are linked to the abundant parting surfaces within shales (i.e. fissility) and the prevalent upper and lower contacts between thick quartzites with under- and overlying-shales (Button and Cawthorn, 2015). The transition from the basaltic Hekpoort Formation rocks (high seismic velocity rocks) to the sedimentary rocks of the Timeball Hill Formation (low seismic velocity rocks) results in the velocity reversal zone observed between the second and third layers, indicated by a solid black line in Figures 9b and 10. `Considered, however, is that the presence of a velocity reversal results in a reduction in the reliability of S-wave velocities of layers located below the reversal zone. Velocity reversal zones reduce the penetration of large wavelengths into the underlying layers, i.e., they act as low-cut filters, thus reducing the mapping resolution and velocity reliability at depth (Groves et al., 2011). Additionally, SW analysis suffers from a reduction in resolution with depth associated with dispersion curve ambiguities at low frequencies. Due to this, not much interpretation is drawn on the exact S-wave velocities observed within the last layer. The prevalence of the velocity reversal zone, throughout the study area suggests that the Hekpoort Formation is present throughout the vicinity of South Deep Gold Mine, contrary to what is observed in the local geological map of the area (Fig. 2a). This is supported by the presence of Hekpoort Formation basalts in areas where Timeball Hill Formation rocks are reported to be outcropping, in the geology map of the mine, see Figure 2 boreholes KMF5, KMF6, and KMF7. However, the confirmation of this interpretation is hampered by the lack of available borehole data around the center and north of the mine. The depth, thickness, and S-wave velocities of the imaged lithologies and corresponding layers are variable, and in some areas, are disturbed by low and high-velocity zones that extend from the near-surface to depths beyond that resolved by the survey (Fig. 11b-d). These high and low-velocity zones are discussed in the following section.



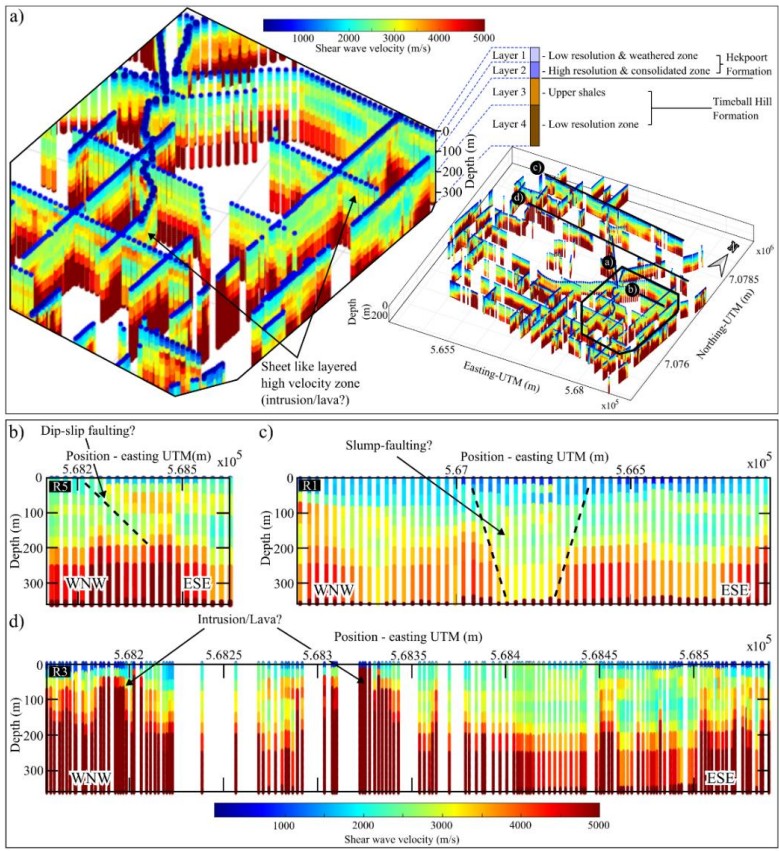

**Figure 11: Local features observed at the site. (a) Sheet-like layering; (b) dip-slip faulting; (c) slump faulting; and (d) Intrusions/lavas. Shown in the 3D grid in (a) are the locations of the zoomed in 3D grid displayed in (a) and the locations of the Vs sections, i.e., R5, R1 and R3 presented in (b), (c) and (d), respectively.**

The low-velocity zones are interpreted as potential dip-slip (Fig. 11b) and slump faults (Fig. 11c). Generally, the low-velocity expression of faults in S-wave velocity models is linked to fracturing and related in-situ weathering that results in the reduction of the shear modulus of the faulted layers. The interpreted dip-slip faulting, in Figure 11b, shows relative vertical displacement between the rocks above and below the fault plane. Meanwhile, the slump-faults, Figure 11c, are largely characterized by wide near-surface footprints that taper down with depth, i.e., they progressively get narrower with depth. In the slump zone, the layers (including the overlaying volcanic rock layers) are observed to be sunk. The identified faulting and possible intrusions are characteristic of the study area and consistent with the findings of Cousins (1962) and Parsons and killick (1990). Cousins (1962) documents that the rocks of the Pretoria Group are displaced by post-Pretoria Group faults that displace both the Pretoria Group rocks and the base of the Transvaal Supergroup, and slump-faulting, which only affects the formations within the Transvaal Supergroup, but not the base of the Transvaal Supergroup. Cousins (1962) interpreted these faults as slump-faulting





originating from the subsidence of the dolomitic units of the Transvaal Supergroup, situated within a formation underlying the Pretoria Group Formations.

The high-velocity zones, extending from depth to the near-surface, are observed in Figure 10c, d and e to be (1) zones of thick lavas of the Hekpoort Formation that could not be resolved well at depth and (2) zones where the dolerite sills that intrude the upper Timeball Hill Formation shales are in close proximity to the Hekpoort Formation lavas at depth.

In the latter case, the velocity high signature appears to be a superposition of both the velocity high signatures of the Hekpoort Formation lavas and the dolerite sills that intruded the Timeball Hill Formation, thus giving an impression of a high velocity that extends to depth and crosscutting the sheet-like layering (Fig. 11d).

Figure 12a-c illustrates the interpolated 3D model view of the grid acquired at the mine and depth slices (i.e., velocity maps) extracted from the 3D model. Figure 12a displays the velocity model of the survey area in 3D. The zones

marked by dashed white and green lines in the vertically sliced 3D model, Figure 12b, illustrate the coherency and depth extend of the low (fracture/fault)- and -high (lavas/intrusion) velocity zones in the area and their interaction with the reversal zone (shown using a black dashed line and situated at a depth of ~ 120 m). The velocity maps in Figure 12c better illustrate the increase of velocities with depth and the presence of low-velocity zones that extend from the near-surface to greater depths.

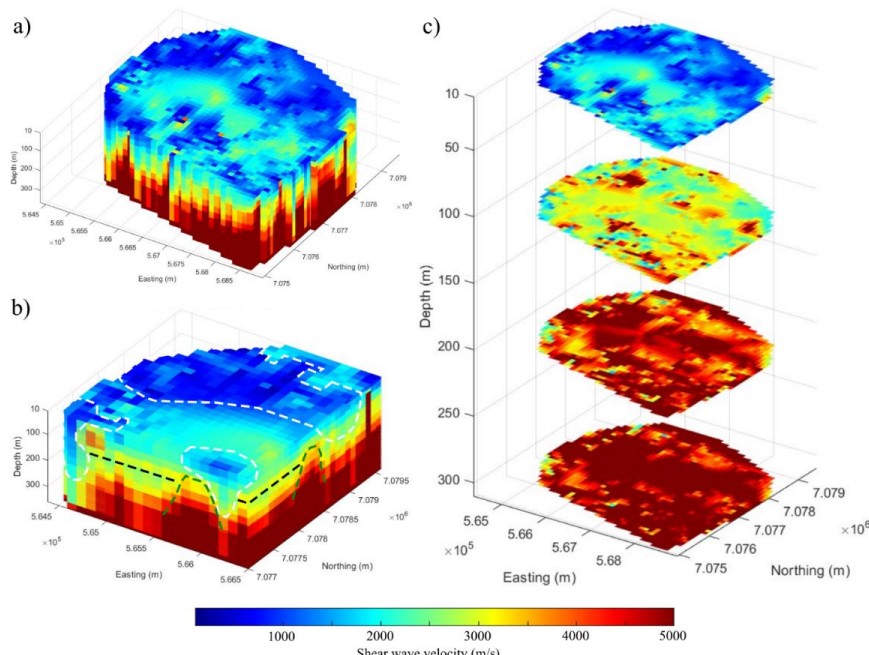

**Figure 12: (a) 3D Vs model of the study area. (b) Cropped 3D Vs model showing the velocity reversal zone (indicated by the black dotted lines), low velocity zones that extend from the surface to greater depths (indicated by the white dotted lines) and high velocity zones that extend from depth to the near-surface (indicated by the green dotted lines). (c) 3D Vs model depth slices illustrating the general variation of Vs velocities with depth.**






### 6.2 Structural linkage between shallow aquifers and deep mining levels

To investigate the structural linkage between shallow aquifers and deep mining levels, we integrated our shallow 3D Vs model with the deep P-wave 3D reflection seismic data acquired in 2003 for mineral exploration, which is used at the mine for mine planning and development. The legacy 3D seismic reflection data survey covers both the South Deep and Kloof mines and was primarily acquired to map the VCR ore body and major fault systems (Manzi et al., 2012). The data generally exhibit good signal-to-noise ratio and show major faults crosscutting the orebody and strong reflections associated with major lithological boundaries from the base (~ 1 km) of the Transvaal Basin down to the West Rand Group below the mining levels (> 3 km). However, the data show poor imaging of the top 500 m due to the sparse seismic survey design (i.e., large spacing between shots and receivers, and between shot and receiver lines). Figure 13 shows a depth-converted P-wave pre-stack time migrated section extracted from the legacy cube with a corresponding pseudo-2D S-wave section present along the line extracted from near-surface 3D S-wave velocity models obtained from 2023 active-source seismic data, respectively. Smoothing, by increasing the gridding bin size, was applied to the presented S-wave section to suppress the more local variations (i.e., subtle variations such as layering) and enhance the visibility of larger (regional) scale trends present in the S-wave data. As stated already, Figure 13 shows that the P-wave seismic section lacks reflectivity in the top ~ 500 m; however, it exhibits strong reflectivity at deeper depths, i.e., between depth intervals of ~ 500 m to 4500 m (Fig. 13a). Overlying the S-wave velocity model on the P-wave section enables near-surface variations and structural complexity resolved by the S-wave velocity model, to be viewed with respect to deeper structures well resolved by the P-wave seismic reflection method, including known water-bearing structures at the mining levels (Fig. 13b). The confirmed water-bearing geological structures at the mining levels include the three north-south trending dykes (here referred to as D-1, D-2 and D-3) that cross-cut the gold orebodies (VCR and Upper Elsburg Reef package) at depths between 2.9 and 3.5 km below the ground surface. The source of the water found at the mining level, along these dykes, is unknown. Furthermore, the continuity of these structures above the mining levels is not known since they are not visible on the current legacy seismic data and not resolved by the underground mapping data, which only focuses on structures within the mining levels. In determining the linkage of these dykes with the overlying aquifers, we analysed the near-surface S-wave velocity zones and investigated their spatial correlation with these dykes. The assumption is that, if the targeted structures intersect the aquifers, then they have the potential to transport water from the overlying Chuniespoort Group dolomite aquifers to the underground workings (Manzi et al., 2012).

Figure 13c and d show near-surface low-velocity zones (indicated by red, green and purple arrows and with velocities varying from ~ 500 m/s to 3500 m/s) that may correspond to known major dykes (D-1, D-2 and D-3) present in the vicinity of the mine and confirmed by underground mapping and drilling. The locations of the low-velocity zones on the S-wave section correlate well with the locations of the dykes, and the consistency of the features and their alignment is observed moving from Figure 15c to d. The low-velocity zones are possibly structurally weak zones that allowed for dyke emplacement (e.g., fracture zones or faults) or are the weathered tops of the dykes. The dykes are mafic in composition, which means that they are less competent in the near-surface compared to silicate rocks, and therefore break down more than the surrounding sedimentary rocks resulting in lower seismic velocities relative to the surrounding sedimentary rocks (Evans et al., 1998). Noted on the seismic sections is that the reflector disturbances





associated with these faults and/or dykes are not visible. This could be because (1), in the case of faults, their throws
are not large enough to be resolved by the seismic data, and (2), in the case of the dykes, they are near-vertical with

thickness less than the resolution limit of the legacy seismic reflection data. These dykes and linked near-surface low-
velocity zones are possible water source conduits that act as pathways for water migration from the Chuniespoort
Group dolomites into the VCR and UER mining levels. Several similar dykes and faults that displace both the BLR
and VCR in the area, but are not observed in the conventional seismic section, are reported in the works of Manzi et
al. (2012).

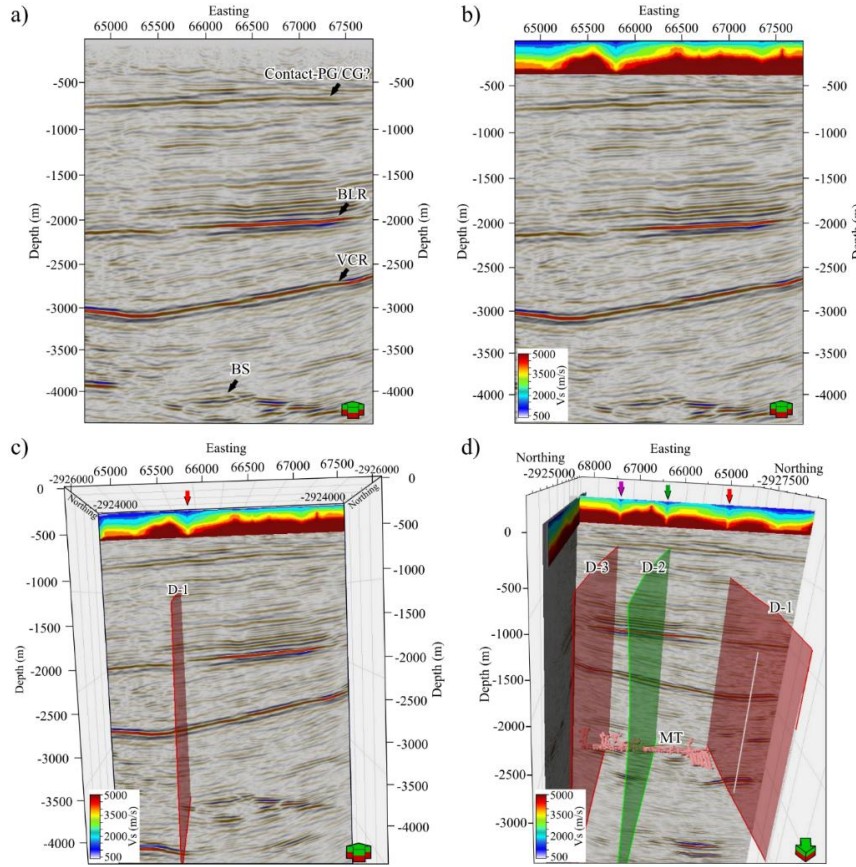


**Figure 13: Comparison between P-wave reflection seismic data and S-wave velocity models. (a) Pre-stack time migrated
seismic section (depth converted) extracted from the 2003 legacy South Deep 3D seismic data. The reflection seismic section
shows strong and continuous reflections related to the contact between the Pretoria and Chuniespoort Groups (Contact-
PG/CG), Black Reef (BLR) unconformity, Ventersdorp Contact Reef (VCR) unconformity, and the Booysens Shale (BS).**
**(b) P-wave reflection seismic section with the S-wave velocity model overlain on it. (c) and (d) show major dykes (D-1, D-2
and D-3) mapped at the mining level, i.e., at a depth of ~ 3500 m, and their possible continuity and propagation to near-
surface rocks (top 300 m) along the fault zones, indicated by the red, green and purple arrows, respectively.**



**7 Discussion**

**7.1 Subsurface conceptual geological model drawn from the integrated datasets**

Overall, the S-wave velocity results attained in this study iterate the advantages and advances that can be made from computing SW analysis on DSR exploration profiles. Comparing the geological information revealed by the S-wave velocity models with proximal borehole data (ground truth information) and known geological characteristics of the area illustrates that the subsurface information inferred from the S-wave velocity models is in agreement with a priori

information. In Figure 14 we provide a geological model that gives a summary of the geological and deformation events present in the near-surface to deeper stratigraphic levels of the study area, moving from the pre-Transvaal to post-Bushveld time period. The geological model focuses on the combined near-surface and deeper P-wave reflection seismic, mine mapping and drilling information. The geological model commences with the deposition of the Central Rand Group of the Witwatersrand Supergroup and the subsequent emplacement of the basaltic lavas of the Ventersdorp

Supergroup (Fig. 14a). This is followed by a pre-Transvaal deformation event that displaces the stratigraphy of both the Central Rand Group and Ventersdorp Supergroup, respectively (Fig. 14a). Subsequently, the Chuniespoort and Pretoria Groups of the Transvaal Supergroup are deposited (Fig. 14a). A post-Transvaal deformation event occurs thereafter, reactivating pre-existing faults, forming faults that displace the entire stratigraphic sequence including the Transvaal Supergroup and consists of associated splay faults (Fig. 14b). The faults act as planes of weaknesses and

serve as a pathway for preferential emplacement of intrusions (doleritic dykes) during pre-, syn-, and post-Bushveld times (Fig. 14c). Doleritic sills are emplaced at an angle to the doleritic dykes along bedding planes of the Pretoria Group rocks (Fig. 14c). Lastly, slump faulting (post-bushveld deformation) occurs due to the dissolution of dolomites of the Chuniespoort Group and displaces the overlying Chuniespoort and Pretoria Group rocks (Fig. 14d).





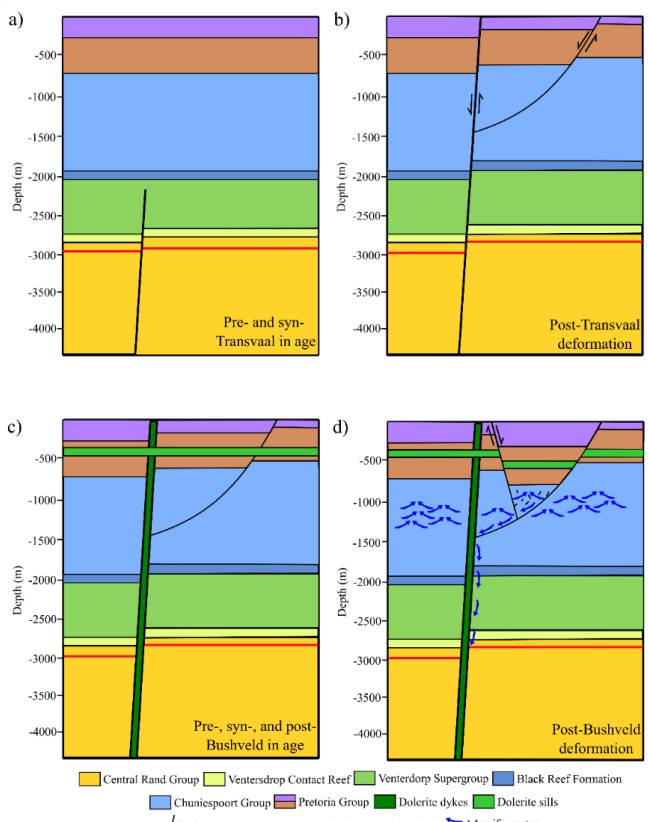

**Figure 14:** Generalized geological model illustrating the sequence of geological and deformation events identified within the study area: (a) deposition of the Central Rand Group and subsequent emplacement of the Ventersdorp Supergroup, followed by a pre-Transvaal deformation event, and thereafter, the deposition of the Transvaal Supergroup (Chuniespoort and Pretoria Groups); (b) faulting associated with a post-Transvaal deformation event occurs reactivating pre-existing faults and displaces the entire stratigraphic sequence; (c) the intrusion of doleritic dykes and sills along planes of weaknesses such as faults and bedding planes (pre-, syn-, and post- Bushveld in age); (d) slump faulting associated with the dissolution of dolomites results in the displacement of the overlying Chuniespoort and Pretoria Group rocks (post-Bushveld deformation). The dolomites of the Chuniespoort Group act as water sources (aquifers) and are displaced by faults and dykes that link them to the Upper Elsburg Reef package (underground workings at depth) and thus, act as pathways for water migration from the near-surface to underground workings (water conduits).

**7.2 Delineating water conduits using shallow and deeper velocity models along with mine mapping and drilling information**

Gold-bearing strata of the West Rand Goldfields are overlain by an approximately 1 km thick Chuniespoort Group dolomite sequence, which is associated with karst systems that host considerable volumes of water (Fig. 14d,



Wolmarans, 1984; Van Niekerk and Van der Walt, 2006). The dolomites (water source) are disturbed by several fracture zones, faults and dykes that link them to the underground workings at depth and act as pathways of water migration from the near-surface to underground workings (water conduits) (Fig. 14d, Manzi et al., 2012). Thus, water inflow in mining districts and water inrush are some of the serious and difficult-to-predict challenges that affect mining

processes and efficiency at the mine. Mining engineers have developed several ways of reducing water inflow in mine workings, however, creating a reduction procedure is difficult without knowing the exact locations of water pathways into the mining levels. In this study, we combine a near-surface S-wave velocity model of the area, computed from the EraMin3 Future Project exploration DSR surveys, with legacy P-wave reflection seismic data available in the study area, and information obtained from underground mine mapping and drilling to (1) characterize and better

understand near-surface lithological variations, and (2) enhance the imaging and delineation of faults and dykes that extend from the near-surface to the deeper mining levels, and that may act as water conduits (Fig. 14d). Generally, it is difficult to exactly determine which faults and dykes act as water conduits, however the dykes integrated in this study have been confirmed through underground mapping to be water-bearing. The results obtained in Figure 13 demonstrate that incorporating a near-surface S-wave velocity model along with deeper P-wave seismic reflection

models, mining level mapping and drilling information enables a more comprehensive, accurate and reliable understanding of the subsurface environment to be drawn. In the case of South Deep Mine, it allowed for possible water conduits linking the Chuniespoort Group dolomites with the deeper situated mining level to be delineated with more certainty (Fig. 14d). From the tectonic point of view, the near-surface S-wave velocity models show that these dykes cross-cut the entire stratigraphy including the Transvaal Supergroup, indicating that these dykes were active

post the deposition of the supergroup. The observation indicates that these dykes are associated with pre-, syn-, and post-Bushveld deformation events and are likely associated with intrusion events such as the Bushveld Complex, Pilanesberg Complex and Umkondo Large Igneous Province.

**8 Conclusion**

SW analysis, conducted on DSR surveys using MASW, was used to investigate the near-surface (top 360 m) at the vicinity of South Deep Gold Mine (South Africa) and investigate the effectiveness of integrating a near-surface S-wave velocity model with a deeper P-wave seismic reflection model, mine mapping and drilling information to aid in delineating possible water conduits. To improve the robustness and spectral resolution of the dispersion images and thus of the S-wave velocity imaging, we were compelled to use large spatial windows to process the data. Additionally,

to improve the data coverage and subsurface mapping in the study area, we utilize the reciprocity principle to increase the data density in areas where no receiver locations, but source locations, were present.

The attained near-surface S-wave velocity models revealed the occurrence of sheet-like layering, and the presence of a S-wave velocity reversal zone, marking the contact between the Hekpoort and Timeball Hill Formations constituting the upper formations in the study area, respectively. The S-wave velocity analysis also revealed the occurrence of

intrusive dykes, dip and slump faulting, which are all geological features characteristic of the West Rand Goldfield.



As the mine is to be extended in the near future, the near-surface results obtained in this research reveal the need for more detailed near-surface investigations to be conducted to avoid placing future mining infrastructure in areas where they may be compromised.

Integrating a near-surface velocity model with a deeper P-wave seismic reflection data, mine mapping and drilling information introduces a novel approach to delineating possible water conduits to deep mining levels and proved to possess great potential for delineating water conduits and possible structures that may pose a risk for deep mining. Of significance is that the results obtained in the study reveal that there is a lot to be gained from conducting SW analysis on DSR exploration surveys. They show that complexities that cannot be resolved by single techniques due to intrinsic limitations such as investigation depth, subsurface complexity, and velocity inversion ambiguities can be better handled by integrating shallow and deep subsurface velocity models and ground truth data.

**Acknowledgements**

This research was funded by FUTURE project (Fiber-optic sensing and UAV-platform techniques for innovative mineral exploration) of the ERA-NET Cofound on Raw Materials (ERA-MIN3), Advanced Orebody Knowledge (AOK), Vinnova (Sweden), Ministry of Universities and Research (Italy), National Agency for Research (France) and DSI-NRF Centre of Excellence (CoE) for Integrated Mineral and Energy Resource Analysis (CIMERA). Much gratitude goes to the Department of Science and Innovation (DSI) for funding the South African partners in the Future project. We would like to extend our gratitude to South Deep Gold Mine of Gold Fields Ltd. for granting us access to their mine, permission to conduct research and publish the outcome of the research. We also wish to extend our gratitude to researchers and postgraduate students from the Wits Seismic Research Centre at the University of the Witwatersrand, Politecnico di Torino, Uppsala University, and Venda University for their contributions to the success of the ERA-MIN3- Future Project.



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
