# Peer review of "Near-surface characterization and delineation of water conduits at South Deep Gold Mine, South Africa"

_EGUsphere, 2025_

## Referee Comment (RC1)

Dear authors and editor,

the presented study is quite interesting and well fit the scope of the issue. The overall idea, set up, acquisition and processing are good and well-presented and the text is mostly well written. Respect to the proposed interpretation and conclusion instead I have some major concerns. The interpretation appears in some aspect too optimistic on trusting the model results and it will benefit of more detailed information regarding the used geological constraints. Part of the conclusions instead are based on a comparison with a legacy data that may not be a proper reference for this study. I would suggest a major revision, not for the amount of needed work, but for the importance of the requested updates that, if not properly addressed, may invalidate the main results and conclusions.

Following you can read my specific comments regarding the main encountered issues and suggestions for more discussions, and technical comments mostly regarding the form of the text.

**Specific comments**

Almost all figures require bigger labels font size, it is too small and difficult to read. In general, many figures are too small and is difficult to see the details that you point out, consider adding zoom windows when needed and insert the figures as big as possible in the manuscript.

Results and interpretation chapter will benefit of a rewording to make the text clearer, sharper and to the point.

In figure 4c it looks like you got some DCs with particularly low phase velocity in P1 respect to the average results, can you map it in the data and in the results?

How confident are you with the velocity reversal? Did you tested if it is data driven or if it is some reminiscence from the starting model? In general, discuss how much the starting model is affecting your results.

The interpretation and results shown in figure 10 present different issues and they need to be updated and revised after consideration of each of them:

(1) Your inverted model seems to be too deep, you said you have a good coverage up to 350 m of wavelength and a reasonable one up to 550 m of wavelength. This means that your maximum reliable depth is approximately 150 to 200 m, without considering its possible reduction related to the velocity reversal and the issues related to the lower resolution in different areas as you state in the text. So, everything deeper than 200 m should be removed and definitely not considered for interpretation (you are interpreting down to 360 m that is almost double the depth). If you have some explanation to justify the reliability at this depth, then clearly state it in the text.

(2) The S-wave velocities in the upper part of the model are mostly reasonable, but 4500-5000 m/s (mostly from the bottom part of the model) are extremely high even for magmatic rocks, and you are attributing them to sedimentary rocks that should have lower ones. You can also see that most of these velocities stop at 5000 m/s indicating that they would go even higher if allowed to. These numbers are not reasonable for what I know and I suspect that they are a result of the starting model, but if you have some particular case in the area that justify these numbers, then state it in the text and indicate some reference. If you have sonic logs from the boreholes it will be very useful to compare them, if they are not available then you should report some velocity from literature for these formations or for similar lithologies.

(3) You talk about an internal distinction in the Hekpoort Formation between weathered and not weathered basalt, do you know if this distinction is reported also in the boreholes? If yes it will be

nice to show in the figure. This contact some time shows a strong velocity contrast while other times it is almost not visible. Is this due to the data quality changes, is it a real feature with internal differences on this contact, or are they different contacts (maybe with the clays reported on top of each borehole)? Discuss it better and use references when possible.

(4) You state that when the basalt is thicker than 200 m, the contact with the Timeball Formation is not visible in the inverted model. I think this is not due to the thickness of the basalt itself but to the depth of the contact. As I said in point (1) of this comment, I will not expect to see anything deeper than 200 m from this data, and therefore if the basalt is already 200 m thick you will simply not see its base contact because you do not have data from it.

(5) what is your expected resolution of the model? what is the minimum thickness you expect to resolve? how much accurate is the location at depth of a contact (5, 10, 20 m)? write it in the text and consider it during your interpretation.

Figure 12 confirms what I said for figure 10, almost everything below 200 m of depth looks unrealistic and a result of the inversion. If this is the case cut the model and show it down to 200 m of depth, you will also have a lower range of velocities that will highlights better your low- and high-velocity features while still keeping most of them.

In the comparison with the 3D seismic legacy data (Figure 13) you state that some of the modeled low-velocity zones are correlated with dikes detected at the mine level (3 km deep).

(1) How constrained are the geometries of these dikes? What is known about them? What is their thickness, dip and strike? Are they constant along the whole length of the dike? Do you have any data showing that they are expected to reach the near surface and do not stop at deeper levels? Do you have a surface expression of them? Just assuming that they move vertically for 3 km is not enough, you need to explain why these geometries are likely to be real and how much confidence we have on them.

(2) Why they are not visible in the seismic sections? You give some possibility but they need to be based on more concrete information. It is ok that vertical and thin structures cannot be imaged, but why the surrounding structures and crossed contacts are not affected and shows linear reflectivity? You may get some strong related diffraction in the unmigrated data, did you check it?

(3) Are this low-velocity zones matching with the dikes strike geometries (if known) also in 3 dimensions? In the figure you show only the crossing with one section and in the text you do not specify it.

(4) You state that the velocity in the dike can be lower than the surrounding sedimentary rocks under weathering conditions. But you never state how deep do you expect this weathering conditions in this area, I assume that at some point the dykes will not be weathered and their seismic velocity will be higher than the sedimentary surrounding rocks. Where is this crossing point? Is it deeper than your model?

In general, to can relate the low-velocity zones with the dikes you have to add more information and constraints about the dikes and where this information come from. Especially when you use them for one of the main conclusions of your work.

I do not understand what is the benefit you get from comparing the SW model with the seismic legacy data. The only deep structure you use in your interpretation are the dykes, that are not imaged, and the geological model, that seems to be derived from regional knowledge and not from the seismic sections. At the shallow depth where you have the SW model the seismic section has no good data available for any comparison. So, what is the scope of comparing them? State it clearer in the text.

You should compare the obtained SW model with the section resulting from the new active seismic data. At least the 2D profiles should have a better quality in the shallow part respect to the legacy 3D seismic data and they will allow you to better constrain and understand your results. You could also try some first break traveltime tomography and compare the two models, low- and high-velocity zones should mostly correspond for P and S waves in these cases.

I like your interpretation in figure 14 and it sounds quite likely, but you need to fix the previous model interpretations and clearly express what you can see from the data, what your results are adding to previous knowledge of the area, and what instead is just a suggestion.

One of your main conclusions is that the faults (together with the dykes) are one of the main pathways for water migration in the studied area and that this information is very important for safe mining operations. I agree with all of this, but I would like to see the whole results from your model and not only 2 examples from figure 11b and c. How many of these faults were you able to detect? Can you see their 3D geometries? Can you map them in the whole area? How deep can you map them? What are the relations between faults and formations seen in the data (and not known from the overall geology)? What are the limitations of the method? Were you able to detect them independently from the data coverage, dip angle and offset? Or there are preferential geometries that are emphasized from the data acquisition (a certain strike and deep for example)? Discuss all of this in the text.

Finally, you are using a new reflection active seismic data (3D and 2D), processing it for SW analysis and comparing with results from an older acquisition with different parameters not better specified, to show that you get information in the shallow area that you do not have in the 3D imaging. I agree with the statement but this comparison is not fair due to the different acquisition and of 20 years of evolution in the seismic industry, instead, you should compare the results from the same acquisition and with the same parameters for a proper comparison. You can then suggest at which depth it is better the SW analysis and at which the seismic reflection imaging. And you should distinguish between the 2D and 3D cases, since they have different resolution and coverage, to distinguish in which cases this method is useful (maybe all of them but you need to show it). At the end, the main question is "Why should someone do SW analysis instead of the more traditional seismic reflection processing, since the acquired data is the same?", it is not faster nor cheaper, so you have to show that you get better results or extra information from the same acquisition.

**Technical comments**

Line 68 – "Malehmir et al, 2015"

Line 98 – Write "(Fig. 1) and not "(Fig. 1a and b)"

Line 98-101 – Rewrite the phrase more clearly, consider to split it in 2 phrases. State clearly that the following phrases are a general setting and later state that you are going into detail of the interested area.

Line 102 – It misses a "and" after the coma.

Line 130 – Remove the symbol "~" from the age since you are already giving an error range. The use of the age unit in "Ga" for line 118 and "Ma" in this and the next line for the same order of magnitude is slightly confusing, I suggest to be consistent all along the text.

Line 132 – "(Dorland, 2004)"

Figure 2 – This figure needs some update. (1) a and b are too small, the text is not readable and the profiles are not distinguishable. (2) the legend in a is only for the geology while the seismic sensors are described in the caption, but then in b the legend is referred to the seismic sensors, and in none you are referring to the boreholes. You may try to combine a and b on a single figure with a single complete legend or remove the satellite view, it is interesting but not of high relevance.

Line 167 – I think you mean "slip-sweep" and not "sleep-weep". Check all along the manuscript.

Line 177 – You can define the distorted layout as crooked or moderately crooked profiles. Specify this especially for the 2D profiles.

Table 1 – Fix slip-sweep and slip time. Only the sweep frequencies for the 3D grid are linear?

Line 186 – Remove "(Fig. 2b; Table 1)". It is just confusing.

Line 192 – "Clearbout and Green, 2008"

Line 195 – "Clearbout and Green, 2008"

Line 206 – "(Socco et al., 2009)" is in between two dots. Remove it or remove the first dot.

Line 228 – "CR gathers" write it as previously in the text (CRGs).

Line 220-230 – You state that the used spatial windows are mostly "relatively large", can you add what would have been the optimal or desired window size for the analyzed datasets?

Line 233 – With the provided figure no ground roll is "clearly visible". The picture is too small and the resolution low, maybe zoom the shot gather in the near offset if you do not need the far offset traces. In general, you do not use the term "clearly", it is not a very scientific term.

Line 234 – "after transforming the data" You may specify "the data in the red dashed rectangle" to do not confuse the reader.

Figure 4 – The figure 4a is too small and low resolution, consider showing only part of it or adding a zoomed window. In figure 4d and e you use the Easting as x axes, but if I understood correctly they represent profile 1 and profile 2 that are perpendicular to each other and none of them parallel to the Easting direction. Did you consider to plot them versus the profile distance or one the easting and the other one the northing instead? It may be more representative of the real situation and easier to read in this case. Unless what you are showing is a projection of the profiles on EW direction, but in this case, you should explain it better in the text (same for figures 9a, 10 and 11).

Line 249 – In figure 5 caption you may add "plotted as a function of wavelength and spatial location" as stated in the caption of figure 4 (remove depth), to keep consistent wording along the manuscript and make it easier for the reader to follow. Make the labels bigger in the figure.

Line 272 – Add a reference or explanation for "(considering approximately depth = wavelength/2.5)."

Figure 7 – Center "Phase velocity (m/s)" label.

I suggest to consider the LCI chapter as a subchapter of data processing (4.1 instead than 5).

Figure 8 – What is the inset shown in 8b? state it in the caption or show it in the figure (or both), I think it is a zoom of the Global misfit but it is not clear.

Line 295 – In this study you used a laterally invariant initial model for the whole area. Since the area is quite big and heterogeneous, did you test how a more detailed initial model affects the results? and more in general, did you tested different initial models to discriminate dependencies of the results from the starting model itself? Discuss it

Line 311 – Fix "Figs. 9a and b)".

Line 312 – What do you mean by "variations in the study area are resolved"? be more specific.

Line 315 – Fix "Figs. 9a and b)".

Line 316 – Remove the dash and use parenthesis instead.

Line 315-320 – Try to be clearer and more concise on what you want to highlight. What do you mean by "low- and high-velocity zones"? It is normal to have low and high velocity unless you are in a homogeneous media. Remove terms as "Evident" and "prominent" and use more scientific and quantitative terms (i.e., in figure 9c is visible a S-wave velocity reversal zone at around xx m of depths).

Line 320 – Remove "(Fig. 9c)" if you already stated "In figure 9c" at the beginning of the phrase.

Figure 9c – The 3D grid plot is very interesting with the full data view, but if you want to use it for highlighting the correlations between datasets it is not useful. The correlations you indicate in the circles are not visible and just take time for the reader to search for them. Or you modify the figure to make these areas more visible (remove some of the data, use zoom in windows, etc.) or you remove it from the figure and just state in the text and the reader will have to trust you. It is also very difficult to understand which data is from the 2D profiles and which one is from the 3D acquisition.

Figure 10 – In the legend write "inferred variations within the formations", using Hekpoort-Timeball Formation is confusing.

Line 398 – How did you do the interpolation? Did you consider the different coverage and the different resolution of the datasets? Write it in the text.

Line 398-404 – Rephrase it to be clearer.

Line 412-419 – Move to the "Site and data" chapter.

Line 424 – If you "already stated" it there is no need to repeat. Give all needed information at the same point.

Figure 13 – You do not state in the caption what MT stands for (I assume mine tunnels?).

Figure 14 – You draw the dyke all the way to the surface, do you have geological evidences of it at the surface?

Line 518 – From what you showed and said so far, the S-wave velocity model does not show any dyke. It shows some low-velocity zone that you suggest may be related to dykes, but you need to provide more information as detailed in the general comments if you want to state this. The low-velocity zones could be just due to faults and local fracturing.

Line 538 – Compromised by what? Specify with some example.

Line 539-541 – This statement is subjected to the replies and additional information you will provide respect to my specific comments.

Line 592 – I cannot find this reference in the text. Check and in case remove it.

Line 604 – This reference is a copy of the one just above, remove it.

Line 639 – I cannot find this reference in the text. Check and in case remove it.

Line 657 – I cannot find this reference in the text. Check and in case remove it.

---

## Referee Comment (RC2)

**Near-surface characterization and delineation of water conduits at South Deep Gold Mine, South Africa**

By Sikelela Gomo, Farbod Khosro Anjom, Chiara Colombero, Mohammadkarim Karimpour, Bibi Ayesha Jogee, Musa S.D. Manzi, Laura V. Socco

The article presents an alternative approach to inferring the potential mining issues at depth by characterising and evaluating near surface properties. Central to this approach is the hypothesis that specific structural features, such as dykes, foster a seismically detectable connection between shallow aquifers and deeper mining zones.

While conceptually compelling and potentially valuable in mine planning and risk assessment, the integration of 3D shear-wave (Vs) velocity models with legacy P-wave reflection data, though innovative, raises questions regarding dataset quality and compatibility.

**Observations/comments:**

Strengths

The study effectively combines newly acquired MASW seismic datasets with legacy P-wave reflection data, demonstrating a creative approach to multi-scale geophysical integration.

The work is clearly motivated by the need to reduce deep mining hazards, by linking geological structures such as dykes and fracture zones to deep mining issues.

The authors succeed in extracting new value from older seismic records, offering novel geological interpretations

The authors demonstrated very competent approach to analysis of surface waves using sub-optimal seismic data. The geophysical investigations are combined with an exhaustive geological literature.

The geophysical results are interpreted against a well-researched and expansive geological framework to strengthen conclusions.

Comments/questions

MASW: Claimed depth of investigation of 360 m seems to be justified with very few data points (Fig. 8).From most of the displays provided the maximum depth appears to be around 200-220m, which is more likely to be the case, based on the acquisition parameters and the spread length over which surface waves could be traced on the low-res images provided.

Fig.8 phase velocities are reasonable. The inverted velocities go to 5000 m/s, seems much to high at depths of 200-300 m. Pretoria complex: Vp (5.0-6.5 km/s), Dolerite intrusions 6.0-6.8 Km/s. Hence Vs higher than 3.8 km/s can hardly be expected.

See for example Altindag, R. (2012), *Correlation between P-Wave Velocity and Some Mechanical Properties*

*And Kgaswane et al., 2012,* Shear wave velocity structure of the Bushveld Complex, South Africa; *Tectonophysics*.

MASW or P and S-waves:

The objective of the article is to document the use of MASW technique to characterise the near surface and connect the anomalies found to the issues encountered at the mining depth level.

Receiver spacing of 10m risks spatial aliasing of surface waves. A frequency–wavenumber (F-K) analysis of both the newly acquired and legacy datasets would significantly strengthen the validation of the applied methods. If remedial steps were taken to mitigate aliasing risks, they should be documented to increase the confidence in the findings.

It remains unclear why P-wave refraction arrivals, which are visibly present in the provided shot records, were not incorporated into the geophysical analysis. Integrating these data with shear-wave profiles would offer a more robust inversion framework. Furthermore, by linking P- and S-wave velocity models, the authors had the opportunity to compute additional elastic parameters, particularly Poisson's ratio, which could enhance aquifer characterization.

Figures and data presentation

Figure 4 shows Rayleigh waves that are barely visible and limited to a spread length of approximately 200 m. Based on standard MASW practices, this propagation range does not support the claimed investigation depth of 360 m, especially when employing 5 Hz geophones and assuming realistic shear wave velocities for the site.

Moreover, the vibroseis source utilized in the survey is known to produce relatively weak surface wave energy. Its long-duration sweep increases the risk of mixed wavefields, which may affect dispersion curve extraction.

Figure 6 exhibits well-developed P-wave refractions extending across the full 2 km spread. Surface waves appear to be present, at best over 200m length, and/or spatially aliased. The small display scale and low resolution of the field data images severely hinder a proper assessment of wavefield characteristics in both Figures 4 and 6.

To allow for accurate interpretation of all wave types, significantly improved display quality is necessary. I recommend that representative shot records from both the new and legacy surveys be shown independently, without additional overlays, so that readers can assess signal quality and wavefield content with greater clarity.

Figure 13 suggests that the legacy data may have undergone excessive (signal-to-noise ratio) SNR enhancement, potentially resulting in an overly smoothed image that masks subsurface discontinuities. If access to the raw legacy dataset is possible, reprocessing it with modern imaging techniques could yield a significantly different structural interpretation. Moreover, the reprocessing may bring more clarity in the near surface, possibly up to 200 m depth.

Lows in Vs at the position of interpreted dykes could be related to the computations since dolerite dykes are likely to have a higher velocity than the surrounding rocks.

Recommendations

The investigative approach is conceptually appealing yet suffers from poor presentation of field data. High-quality visual displays are crucial for transparent interpretation. This needs to be corrected.

I believe that the analysis would be strengthened by incorporating P-wave refraction tomography, given that refraction images already span the full spread length.

F–K (frequency–wavenumber) plots should be included to assess the integrity of surface wave sampling and verify spatial aliasing.

The stated depth of investigation needs more rigorous justification. I recommend including a brief overview of MASW methodology and the commonly accepted rule of thumb for depth estimation.

The geology chapter could be much more concise and related to the analysis shown. Geological referencing is way to extensive for a little benefit to the reader.

---

## Author Comment (AC1)

Dear Samuel Zappalà,

Thanks for the careful review of our manuscript. We hereby provide a point-to-point response to all your remarks. Our comments are outlined in blue.

Kind regards,

Sikelela Gomo                                                                 Johannesburg, 14/09/2025

(corresponding Author)

RESPONSE

The presented study is quite interesting and well fit the scope of the issue. The overall idea, set up, acquisition and processing are good and well-presented and the text is mostly well written. Respect to the proposed interpretation and conclusion instead I have some major concerns. The interpretation appears in some aspect too optimistic on trusting the model results and it will benefit of more detailed information regarding the used geological constraints. Part of the conclusions instead are based on a comparison with a legacy data that may not be a proper reference for this study. I would suggest a major revision, not for the amount of needed work, but for the importance of the requested updates that, if not properly addressed, may invalidate the main results and conclusions.

We thank the reviewer for the comment, especially that the study is interesting and fits the scope of this special issue. With respect to the concerns raised regarding the interpretation of the results and conclusions drawn, we realise that we might not have provided enough information in the introduction regarding the mining-and research-curiosity driven studies that have been conducted and concluded at South Deep mine to highlight the current knowledge gaps and shortcomings in the existing mine geological models and tectonic models (as derived from legacy borehole data, seismic data, surface and underground mapping data, geochronological data, etc.) to motivate better the primary aim of characterizing the near-surface geology using the surface wave analysis. We have added a few lines in the introduction and throughout the paper to emphasize these points, especially the importance of utilizing the legacy data in this study. To provide readers and reviewers with a broader context, we have prepared a document that addresses some of the comments raised here and also highlights the research work conducted at the site from which the interpretations and conclusions were drawn. Summarily, the same legacy 3D data have also been used by Manzi et al. (2012b) to map potential conduits of water and methane in the deep gold mines (including the South Deep mine). It has been used in the application of 3D seismic technique in evaluation of ore resources (Manzi et al. (2012a), and for mapping the distribution and timing of mesoto mega-scale structures and provide constraints on the ore genetic models (Malehmir et al., 2013). The data have been used to image the seismogenic

faults and dykes (Masethe et 2023). The data were used to study the tectonic model of the Witwatersrand goldfields (covering the South Deep mine), focusing on structural setting, mainly the first-order scale structures, but also their associated second- and third-order faults and folds, as well as to constrain the relative chronology of tectonic events. In all the above applications, the results were integrated with borehole data and underground maps to validate the seismic interpretations. In this study, the surface wave analysis (main topic for this manuscript) characterizes the near-surface geological structures, which are not well defined in the previous studies by the legacy data (as discussed in the paper) – the information is new. The near-surface Vs model is then integrated with the legacy data to investigate the structural linkage between the near-surface groundwater aquifer system (within the top 300 m) and the deep mining level (~ 3 km below ground surface). As for the legacy data, they are published and available to authors who have worked on them for the past 15 years. We do not see any reason why we should not use them and incorporate them into the interpretation of the structures that connect the deep mining level and near-surface geology, especially since they are the only current reliable data that map the geology at high resolution from 500 m to 6000 m (or beyond) below the ground surface. The other new datasets (e.g., under Future project) are limited in penetration depth because of the type of energy sources used (1 x 6 ton Minivib in new data vs 35 ton vibroseis trucks (4 fleet) in legacy survey) and seismic survey design (e.g., small offset in new data vs large offsets in the legacy survey), amongst other parameters.

In the sections below, we address other comments one by one:

Specific comments

Almost all figures require bigger labels font size, it is too small and difficult to read. In general, many figures are too small and is difficult to see the details that you point out, consider adding zoom windows when needed and insert the figures as big as possible in the manuscript.

Figures have already been sized as big as possible. The dataset is large and often many subplots are needed. We have improved the resolution of all the images in the manuscript, separated some images and enlarged the font sizes.

Results and interpretation chapter will benefit of a rewording to make the text clearer, sharper and to the point.

The authors of this manuscript are, in large part, English native speakers. This comment is very generic, and we are not sure which parts are, according to you, unclear or not to the point. In any case, based on other comments from other reviewers, the interpretation has been modified in some parts.

In figure 4c it looks like you got some DCs with particularly low phase velocity in P1 respect to the average results, can you map it in the data and in the results?

These are very few very low velocity curves out of 2557 curves, the majority of which consisted of wavelengths depths smaller than the minimum wavelength recorded in most of the data (i.e., 20 m), as evident in Figure 4c P1. These low velocity curves are due to propagation in a very shallow layer of weathered material, which is mostly not mapped by the dataset. In the case of a very high velocity site, such as the one investigated in this manuscript, a few meters of loose material may sometimes generate a low velocity mode that coexists with a higher velocity mode due to the propagation at the interface between the loose layer and the hard rock.

The presented data were acquired using a receiver spacing of 10 m along the high-resolution 2D lines, 25 m along the 3D receiver lines, and 12.5 m for the CRGs computed along the 3D grid source lines. Due to the possibility of picking dispersion curves beyond the Nyquist wavenumber (see Socco and Strobbia 2004 for details about unwrapping aliased portion of the spectra and see also comment to Reviewer 2 for an example), these numbers correspond to the expected minimum wavelengths retrieved in the data. Apparent in Figure 4 is that all the low phase velocities have a wavelength of 15 m and less, which is lower than the minimum wavelength (about 20 m) retrieved in the rest of the analyzed data, thus we did not focus on them. The few low velocity curves have very limited effect on the final results due to the limited penetration depth. We do not think that they have a specific relevance to the final velocity model or that deserve specific attention in the final model. Removing them from the dataset would not change the interpretation of the results.

Socco, L.V., C. Strobbia, 2004, Surface-wave method for near-surface characterization: a tutorial: Near Surface Geophysics 2,4 165-185 https://doi.org/10.3997/1873-0604.2004015.

How confident are you with the velocity reversal? Did you tested if it is data driven or if it is some reminiscence from the starting model? In general, discuss how much the starting model is affecting your results.

This remark is not clear to us. The initial model is reported in Table 3 and does not contain any velocity reversal. All the local 1D models make use of the same starting model that is a growing

velocity gradient. No a priori information about low velocity layer was introduced and this result is purely a result of the data fitting in the inversion. Since it is consistent with the expected geological model, we have no reason to doubt about it. Also, the spatial consistency of the low velocity layer is a further element that supports the presence of the low velocity layer. As explained in the text, we introduced spatial constraints, but the spatial consistency of the low velocity model was evident also in the unconstrained inversion (even though more noisy). Moreover, the depth range where the velocity inversion is present is the one with better coverage of the data (see Figure 7 of the manuscript) and patterns in agreement with a velocity inversion can be observed in the data (see the spectra in Figure 6 of the manuscript). The model parameters have been significantly updated during the inversion, moving quite away from the initial model and evidencing significant lateral variations which are not present in the initial model (1D). For all the aforementioned reasons, we think the low velocity layer is fully reliable. We have added a sentence in the new version of the manuscript to specify all these aspects in case other readers may have doubts about the initial model bias.

The interpretation and results shown in figure 10 present different issues and they need to be updated and revised after consideration of each of them:

(1) Your inverted model seems to be too deep, you said you have a good coverage up to 350 m of wavelength and a reasonable one up to 550 m of wavelength. This means that your maximum reliable depth is approximately 150 to 200 m, without considering its possible reduction related to the velocity reversal and the issues related to the lower resolution in different areas as you state in the text. So, everything deeper than 200 m should be removed and definitely not considered for interpretation (you are interpreting down to 360 m that is almost double the depth). If you have some explanation to justify the reliability at this depth, then clearly state it in the text.

The idea that the investigation depth is about 1/2 – 1/3 of the wavelength is based on an oversimplification of the Rayleigh wave propagation. This limitation is a rule of thumb that is sometimes adopted, but many publications have shown the possibility of extending the investigation depth well beyond this rule-of-thumb limitation. Literature is rich of examples of global search method inversion that have extended well beyond this limit and that have shown sensitivity to the model parameters at depth larger than 1/2 of the wavelength. See for instance: Cao et al, 2020, (10.1190/geo2018-0562.1). Comina et al, 2022, (https://doi.org/10.1016/j.soildyn.2022.107262) used Monte Carlo inversion on more than 50 datasets to statistically show that the relationship between the wavelength and the corresponding investigation depth leads to reliable investigation depth that

Field Code Changed

goes well beyond ½ of the maximum wavelength. There is a significant number of dispersion curves with wavelength beyond 360 m, which we assumed as bottom of our model. This information is clearly visible also in plot e) and f) of figure 7. Moreover, by looking at the obtained model (Figure 10) it is easy to notice that there are several zones in the investigated area where the deeper layers in the model have been updated with respect to the initial model value, thus confirming that the data are sensitive to the model at the selected investigation depth. Here below we also show a sensitivity analysis based on a Montecarlo inversion, for an example dispersion curve to show that sensitivity to model parameter exists beyond 200 m (Figure 1). The computation of the wavelength-Depth relationship (Socco et al., 2017) also confirm that the investigation depth is much larger than the limit you suggest. Moreover, we have adopted a spatially constrained inversion approach. The advantage of this approach is that, for those zones where information is not available in a certain wavelength range, the spatial constraints will propagate the information of the zone where the data are available. This certainly means that for the dispersion curves that have no penetration, the deep portion of the model is a sort of interpolation of the surrounding 1D models.

To avoid speculations on information with poor lateral resolution we have cut our models to 300 m. This change in the representation of the velocity models does not affect the interpretation that is mostly focused on the shallower part of the model.

[Figure]

Figure 1: Sensitivity analysis of the medium-frequency band dispersion curve from the dataset (Line P1) to velocity variations in the deeper layers. (a–b) Selected dispersion curves and corresponding VS models from 1D Monte Carlo inversion (Socco and Boiero, 2008). (c) W/D analysis (Socco et al., 2017) indicating the investigation depth as a function of wavelength. (d–f) Sensitivity of the dispersion curve to a 30% perturbation of VS in the half-space. (g–i) Sensitivity to a 30% perturbation of VS in the layer above the half-space.

Socco L.V., C. Comina, F. Khosro Anjom, 2017, Time-average velocity estimation through surface-wave analysis: Part 1 — S-wave velocity: GEOPHYSICS, 82, 3, U49–U59, https://doi.org/10.1190/GEO2016-0367.1

Socco, L.V, D. Boiero, 2008. Improved Monte Carlo Inversion of Surface Wave Data: Geophysical Prospecting, 56, 357-371 https://doi.org/10.1111/j.1365-2478.2007.00678.x

(2) The S-wave velocities in the upper part of the model are mostly reasonable, but 4500-5000 m/s (mostly from the bottom part of the model) are extremely high even for magmatic rocks, and you are attributing them to sedimentary rocks that should have lower ones. You can also see that most of these velocities stop at 5000 m/s indicating that they would go even higher if allowed to. These numbers

are not reasonable for what I know and I suspect that they are a result of the starting model, but if you have some particular case in the area that justify these numbers, then state it in the text and indicate some reference. If you have sonic logs from the boreholes it will be very useful to compare them, if they are not available then you should report some velocity from literature for these formations or for similar lithologies.

We do not understand the remarks of velocity going higher if they "would be allowed to". The data fitting is equally good along the whole frequency band (see Figure 8a) and we do not see any reason why the velocity should be underestimated. As it can be noticed in table 3, the maximum velocity in the initial model is 3000 m/s so we do not see why the reviewer think that the inversion results are biased by the initial model.

Concerning the very high values of the velocity, we have thoroughly analyzed the results and in fact the value of 5000 m/s is associated to the deeper layer of the model which is the one that presents the lower sensitivity. We have hence changed our data representation and limit our representation to 4000 m/s. In Figure 2 we show some examples of lines with the new color scale, and we show that this representation allows the investigation depth of the different portions of the model to be outlined. This change in the data presentation does not change the velocity model in the upper portion of the model on which our interpretation is based.

[Figure]

Figure 2. Vs profiles extracted from the 3D grid. The profiles are limited to shear Vs of 4000 m/s and illustrate the variable nature of the depth penetration of the data.

(3) You talk about an internal distinction in the Hekpoort Formation between weathered and not weathered basalt, do you know if this distinction is reported also in the boreholes? If yes it will be nice to show in the figure. This contact some time shows a strong velocity contrast while other times it is almost not visible. Is this due to the data quality changes, is it a real feature with internal differences on this contact, or are they different contacts (maybe with the clays reported on top of each borehole)? Discuss it better and use references when possible.

We checked the boreholes and spoke with the mine geologists about the descriptions of the rocks in their surface boreholes. The details about weathering and types of basaltic rocks or lavas were not given on the log sheet. The authors have worked with the mine for the past 15 years, and our general observation is that the mine geologists know that the orebody is situated at 2.7 and 3.5 km below

ground surface, so when they log the surface core they are only interested on the lithological contacts and thicknesses of each layer, marking the top and bottom of the individual layer. They are also interested in logging the major fault zones. Details such as the degree of weathering of the rock layer is not of interest to them as it is not associated to mineralization. However, at South Deep mine, the orebody is located at the base of the overlying basaltic lavas (lava-conglomerate contact) at approximately 3 km. In this case, mine geologists, when logging basalts closer to the mining horizon or mineralization, need to differentiate between competent and weathered basalts because the degree of weathering has implications for mine design and hangingwall support, as weathering affects the strength and stiffness of the rock. The current PhD student at Wits University is conducting research on the physical properties of underground borehole samples, including basaltic lavas from the Ventersdorp Supergroup (Plaatjie et al., under review). In this study, the lava samples exhibit Vp in the range between 5915 and 6829 m/s and bulk densities in the range between 2.75 and 2.9 g/cm$^3$. The variability in physical property measurements of the basalts is consistent with our observations of basalts from other neighboring mines (Molezzi et al., 2019; Nkosi et al., 2023). Based on these examples, and the authors' experience with the rocks from the site, we expect the Vs for the lavas to be variable based on the conditions described below.

Regarding a lack of contrast in the Vs across the weathered and not weathered lavas:

This is likely due to weathering (or alteration) and the degree of alteration, which is expected in this study and has been reported in several studies in this region (Mutshafa et al., 2023). For example, in cases where the contrast is not significant, it could be that the other lava layer is less weathered/altered than the other, i.e., the degree of weathering is not huge, hence similar Vs values. In cases where the weathering on the other layer is significant, then we expect significant Vs contrast. The presence of groundwater aquifers and fracture networks plays a role in the weathering or alteration process, which can explain the change of Vs across the section. Ndamulelo et al. (2023) report that weathering of basalts, sills, dykes, and dolomites in the region is highly variable and can extend to considerable depths ($\sim$ 500 m). A change in Vs could also be due to faulting and fracturing (either natural or mining-induced) as already discussed in the paper.

"Mutshafa, N., Manzi, M.S., Westgate, M., James, I., Brodic, B., Bourdeau, J.E., Durrheim, R.J. and Linzer, L., 2023. Seismic imaging of the gold deposit and geological structures through reprocessing of legacy seismic profiles near Kloof–Driefontein Complex East Mine, South Africa. *Geophysical Prospecting*, *71*(7 Special Issue: Mineral Exploration and Mining Geophysics), pp.1181-1196."

"Reczko, B.F.F., 1994. The geochemistry of the sedimentary rocks of the Pretoria Group, Transvaal Sequence. University of Pretoria (South Africa)."

(4) You state that when the basalt is thicker than 200 m, the contact with the Timeball Formation is not visible in the inverted model. I think this is not due to the thickness of the basalt itself but to the depth of the contact. As I said in point (1) of this comment, I will not expect to see anything deeper than 200 m from this data, and therefore if the basalt is already 200 m thick you will simply not see its base contact because you do not have data from it.

As stated above we humbly disagree with this comment related to the investigation depth.

(5) what is your expected resolution of the model? what is the minimum thickness you expect to resolve? how much accurate is the location at depth of a contact (5, 10, 20 m)? write it in the text and consider it during your interpretation.

As it can be noticed in table 3, the minimum layer thickness of our starting model is 30 m. during the inversion process, the layer thicknesses, and therefore the position of the interfaces, have been updated. There is an obvious reduction of vertical and lateral resolution with depth and the resolution of the interface estimation depends also on the velocity and on the local quality of the data. It is therefore not scientifically sound to identify a vertical resolution value that holds for the whole 3D velocity model. Anyway, the minimum thickness of a layer that was produced by the inversion process is ~40 m. And we think this is a reasonable estimate of the vertical resolution.

Figure 12 confirms what I said for figure 10, almost everything below 200 m of depth looks unrealistic and a result of the inversion. If this is the case cut the model and show it down to 200 m of depth, you will also have a lower range of velocities that will highlights better your low- and high-velocity features while still keeping most of them.

Please, see comments above.

In the comparison with the 3D seismic legacy data (Figure 13) you state that some of the modeled low-velocity zones are correlated with dikes detected at the mine level (3 km deep).

(1) How constrained are the geometries of these dikes? What is known about them? What is their thickness, dip and strike? Are they constant along the whole length of the dike? Do you have any data showing that they are expected to reach the near surface and do not stop at deeper levels? Do you have a surface expression of them? Just assuming that they move vertically for 3 km is not enough, you need to explain why these geometries are likely to be real and how much confidence we have on them.

What the authors know about the dykes:

(1) In general, dykes at South Deep mine strike North-South, ~ 20 – 60 m thick, and near-vertical (dip ~ 89-90 degrees). These are reported in papers by Manzi et al. (2012) and Manzi et al. (2013), among others.

(2) The dykes have been confirmed through underground drilling and mapping. Their strike, dip and thickness have been derived from underground mapping and drilling. The information is available in internal mine reports.

(3) The dykes are exposed on the underground mine face. The authors of this paper have seen and sampled the dykes for physical property studies and geochemical analysis as part of the Eramin Future project. One of the seismic profiles (e.g., using DAS) conducted along the tunnel at the South Deep mine, as part of the Eramin Future Project, crosscut one of the dykes presented in this paper.

(4) The dykes presented in this paper are known to host large seismic events, and their orientation and locations have been confirmed by analyzing seismic events along these dykes as part of the in-mine seismic monitoring networks. Their orientations have been confirmed through Focal Mechanism Solutions or Fault Plane Solutions. The information is not yet published, but it is available in the mine reports.

(5) In general, the dykes at South Deep mine (and in the Witwatersrand gold fields) are associated with faulting, meaning that they intruded in the fault zone (which is common for dykes). In the legacy data, it has been found several times that when mine-modelled dykes are integrated with the legacy seismic data, they seem to be located along the fault zones. The authors have worked on several sites in the Witwatersrand goldfields, and this observation has been consistent (e.g., Mngadi et al., 2025). In South Deep, the dykes are often, but not always, associated with faulting.

What is not known is the age of these dykes – an explanation is given below.

(6) South Deep dykes intruded at different geological ages and have been subjected to several investigations (as mentioned above). The dykes are of various ages, such as known post-Karoo

dykes of pre-Cretaceous and Cretaceous age (145 - 66 Ma), Karoo dykes (150 Ma), Pilanesberg (1.30 Ga), Vredefort meteorite impact event (2.05 Ga), the Bushveld Igneous Complex magmatism (2.03 Ga), Transvaal (2.20 Ga), and Ventersdorp (2.60 Ga) (Frimmel, 2014; Manzi et al., 2017; Fuchs et al., 2016; Frimmel & Nwaila, 2020). The post-Transvaal tectonic events that led to the reactivation of pre-existing faults, which are clearly imaged in the legacy seismic data covering South Deep mine and other neighbouring mines (Manzi et al., 2013; Nwaila et al., 2020a), might have created weaker zones for the dykes to intrude and crosscut the Transvaal Supergroup. Movement along faults (naturally or by mining activities) inevitably results in the creation of numerous microfractures, which can be observed and mapped underground (if exposed). Current mining activities can also reactivate dykes (Masethe et al., 2023), promoting water infiltration, movement, and circulation. The South Deep dykes have not been dated, so their age is unknown and cannot be confidently assigned to the specific tectonic event. If the South Deep dykes presented in the paper are related to any of the above post-Transvaal tectonic events, then surely they should crosscut the Transvaal Supergroup and intersect the aquifers. If the dykes do not breach the base of the Transvaal Supergroup, then they are likely to be associated with the dyke swarm from the Ventersdorp-age dykes. The alternative way to understand their timing of activity is to investigate their relationship with the overlying near-surface geology, which is one of the objectives of this study.

(7) Based on the above, the authors have evidence that these dykes exist. Based on the work already published, the authors interpreted these dykes to be associated with post-Black Reef events (like many dykes in the Witwatersrand goldfields) because of the spatial correlation between the Vs anomalies and these underground mapped geological structures. However, we also don't rule out the possibility that the dykes could be Ventersdorp in age, meaning they don't breach the base of the Transvaal supergroup.  Unless the dykes are dated, we will never know their age, but this doesn't stop us from constraining their timing of activity using several datasets at our disposal, including the data presented in this study, to support the interpretation.

(8) There are several dolerite dykes' outcrops at South Deep mine as seen on the surface map (in Figure 2). Some dykes outcrops were observed during data acquisition and sampled for other geological analysis.

(9) Regarding the dyke vertical extent: dykes in general, and at South Deep, are typically near-vertical in orientation and it is not farfetched to assume that they can move vertically for 3 km as some can extend for several kilometers below the surface (Cañón-Tapia, 2008).

"https://www.goldfields.com/reports/annual_report_2016/minerals/reg-africa-south-geology.php#:~:text=Different%20populations%20of%20dykes%20also,north%2Dsouth%20through%20the%20property."

"Cañón-Tapia, E., 2008. How deep can be a dyke?. *Journal of Volcanology and Geothermal Research*, *171*(3-4), pp.215-228."

(2) Why they are not visible in the seismic sections? You give some possibility but they need to be based on more concrete information. It is ok that vertical and thin structures cannot be imaged, but why the surrounding structures and crossed contacts are not affected and shows linear reflectivity? You may get some strong related diffraction in the unmigrated data, did you check it?

Please see the comments above; we believe we have given more than enough information to support our interpretation of these dykes. We have also added more details in the revised manuscript to enhance the understanding of the dykes. The reviewer states, "It is ok that vertical and thin structures cannot be imaged, but why are the surrounding structures and crossed contacts not affected and show linear reflectivity?"

It is not scientifically correct to say that the dykes cannot be imaged because they are vertical and thin, it depends on the seismic survey design, data frequency content, and seismic imaging and seismic interpretation approaches used. In this particular data (legacy 2003 seismic data), the dykes were not directly delineated because of the resolution of the data, which is already discussed in this paper and several papers (Manzi et al., 2012a,b; 2013a,b; Nwaila 2022), and the imaging approaches used during processing (conventional prestack and postack migration approaches using Kirchoff migration). The re-processing of these data using depth imaging approaches is investigated by one of the PhD students, and the results will be reported in the future. The current PhD student is also exploring the diffraction imaging approach on this legacy data to improve the imaging of other complex geological structures. We believe that with more information from the mine, we can also update the velocity model and improve the resolution through depth imaging. However, we are also aware that the data at the current state are one of the most high-resolution data ever acquired in the hardrock environment worldwide. The lack of a good signal-to-noise ratio in the near-surface is probably not due to processing methods, but rather to the seismic design, which focused primarily on

imaging deeper targets for mine planning and development, or a combination of factors (including bandwidth-limited sources and sensors used).

In terms, of the surrounding layers being imaged:
We don't understand this comment, but other layers are probably imaged because they are not near-vertical and there is significant contrast between these lithological units. If the reviewer is referring to the near-vertical structures, such as faults, these are imaged because they crosscut and displace the horizontal reflections with offsets above the resolution limit. As discussed in the manuscript, it is possible that these dykes intruded into the fault zones that have throws below the seismic resolution, hence they are not visible on the legacy data, but imaged by the Vs model.

(3) Are this low-velocity zones matching with the dikes strike geometries (if known) also in 3 dimensions? In the figure you show only the crossing with one section and in the text you do not specify it.

Figure 14c and d show the spatial and vertical extension of the dyke D-1 and its possible associated low velocity anomaly. We investigated the 3D Vs model; these observations were consistent across the 3D model. For visualization purposes, we decided only to show the arbitrary lines across the sections of interest.

(4) You state that the velocity in the dike can be lower than the surrounding sedimentary rocks under weathering conditions. But you never state how deep do you expect this weathering conditions in this area, I assume that at some point the dykes will not be weathered and their seismic velocity will be higher than the sedimentary surrounding rocks. Where is this crossing point? Is it deeper than your model?

The chemical weathering or alterations are not depth-limited in this case study. Weathered dykes and basaltic lavas have been observed at the mining level (~ 3 km) where they are exposed. Although weathering is not reported in these boreholes, weathering can also occur in the top 500 m as demonstrated by Mutshafa et al. (2023), and it is possible to have weathering in certain zones of the dykes along the entire length of the dykes for several kilometres. The main drivers are fluid, fracturing, chemical composition, and other factors discussed above. At South Deep, we have underground water from the near-surface up to 3 km (mining level) and beyond 3 km (since some of the water has residence times in the billions of years, meaning it is sourced from depth, not from near

surface). Mutshafa et al. (2023) report that weathering in the region is highly variable and can extend to considerable depths. They show that weathering in the area can reach depths exceeding 500 m, which is significantly higher than the mapping limits of the computed shear wave velocity model. Additionally, two other factors complicating the interpretation are: (1) weathering of dykes and surrounding country rocks can vary substantially within an area, and (2) the resolution of shear wave velocity models decreases with depth. These factors make it challenging to address the issue of the turning point of the dyke weathering and related velocities. This is impossible to investigate with any surface-based method. The only, but most expensive, way to explore this is to drill through the dyke from the surface (if exposed) to a depth of 3 km and study the variability and degree of weathering.

The following has been added to the manuscript:
"The reduction in S-wave velocities within fault zones is attributed to increased fracturing, porosity, and fluid content, which lead to the physical breakdown (weathering and alteration) of the primary minerals in the in-situ rock into less dense secondary phases. This transformation reduces the rock's elastic modulus, ultimately lowering seismic velocities. According to Mutshafa et al. (2023), weathering in the study area's rocks is highly variable and can extend to depths greater than 500 meters, which is consistent with the presented findings."

"Mutshafa, N., Manzi, M.S., Westgate, M., James, I., Brodic, B., Bourdeau, J.E., Durrheim, R.J. and Linzer, L., 2023. Seismic imaging of the gold deposit and geological structures through reprocessing of legacy seismic profiles near Kloof–Driefontein Complex East Mine, South Africa. *Geophysical Prospecting*, *71*(7 Special Issue: Mineral Exploration and Mining Geophysics), pp.1181-1196."

In general, to can relate the low-velocity zones with the dikes you have to add more information and constraints about the dikes and where this information come from. Especially when you use them for one of the main conclusions of your work.

The low velocity zones can be attributed to both faults and dykes, which are both possible causes of the observed near-surface low velocity zones as stated in the paper (see below)
"The low-velocity zones are possibly structurally weak zones that allowed for dyke emplacement (e.g., fracture zones or faults) or are the weathered tops of the dykes. The dykes are mafic in composition, which means that they are less competent in the near-surface compared to silicate rocks, and therefore break down more than the surrounding sedimentary rocks resulting in lower seismic velocities relative to the surrounding sedimentary rocks (Evans et al., 1998). Noted on the seismic

sections is that the reflector disturbances associated with these faults and/or dykes are not visible. This could be because (1), in the case of faults, their throws are not large enough to be resolved by the seismic data, and (2), in the case of the dykes, they are near-vertical with thickness less than the resolution limit of the legacy seismic reflection data"

Generally the dykes are in close proximity to fractures and fault zones, which correlate with the low-velocity zones. These fractures and fault zones acted as planes of weaknesses for preferential emplacement of the dykes. The dykes are most likely mafic in composition and are thus, less resistant to breakdown (weathering and alteration) compared to silicate rocks. In addition, the dykes that occur in the near-surface are more prone to weathering, weakening their original physical properties. A section on weathering in the study area has been included (see previous comment).

I do not understand what is the benefit you get from comparing the SW model with the seismic legacy data. The only deep structure you use in your interpretation are the dykes, that are not imaged, and the geological model, that seems to be derived from regional knowledge and not from the seismic sections. At the shallow depth where you have the SW model the seismic section has no good data available for any comparison. So, what is the scope of comparing them? State it clearer in the text.

As already mentioned, the seismic survey at South Deep was not designed to characterize the near-surface groundwater aquifer system or near-surface geology. The legacy seismic data, in their current form, provide high-resolution imaging of geological structures from approximately 400 m to 6,000 m depth below the ground surface. The approach of surface wave analysis presented in this paper is intended to supplement the legacy reflection seismic data, borehole data, and mine geological model, rather than compare or replace these methods – it will not be a good approach to compare them, as they are different datasets designed and implemented at the site for different research and mining purposes. The scientific question we are attempting to address by integrating these methods is simple: Is there a connection between the near-surface aquifers and mining horizon levels at the South Deep mine study area? Based on the VS model presented in this manuscript, in conjunction with legacy data and other datasets, our interpretation is that the two systems may be connected at the South Deep mine. The geological model presented here is constructed by integrating all available data, including the seismic data, near-surface velocity model, mining and drilling records, and regional geological knowledge of the study area. The seismic section helps identify broader geological variations at depth, while the shear wave velocity model captures near-surface heterogeneities. Mining and drilling data, along with regional geological insights, are used to incorporate features that may not be imaged by

the legacy seismic (for the reasons already discussed) datasets but are known to exist within the study area. For more information on this discussion, we have also provided an additional document to the Editor to address this comment and others.

A dedicated section on this is present in the discussion section of the manuscript, i.e., see section 5.2

You should compare the obtained SW model with the section resulting from the new active seismic data. At least the 2D profiles should have a better quality in the shallow part respect to the legacy 3D seismic data and they will allow you to better constrain and understand your results. You could also try some first break traveltime tomography and compare the two models, low- and high-velocity zones should mostly correspond for P and S waves in these cases.

We thank the reviewer for his suggestions. The new data are being processed by a PhD student and the final output will be compared with the SW model and the legacy seismic data. Also, comparing the P-wave first break travel times from new data and the Vs model is a good idea that we will incorporate in the future.

I like your interpretation in figure 14 and it sounds quite likely, but you need to fix the previous model interpretations and clearly express what you can see from the data, what your results are adding to previous knowledge of the area, and what instead is just a suggestion.

We do not understand what the reviewer means when he distinguishes between "what you can see" and "suggestion". The interpretation of the data is based on the knowledge of the site by some of the authors, based on decades of work at the site. We therefore value their interpretation strongly. Interpretation is always based on experimental evidence and knowledge and it provides a model. The model is what the authors consider the most plausible explanation of the data based on all the available information so we are not sure what the reviewer means with this remark.

One of your main conclusions is that the faults (together with the dykes) are one of the main pathways for water migration in the studied area and that this information is very important for safe mining operations. I agree with all of this, but I would like to see the whole results from your model and not only 2 examples from figure 11b and c. How many of these faults were you able to detect? Can you see their 3D geometries? Can you map them in the whole area? How deep can you map them? What are the relations between faults and formations seen in the data (and not known from the overall geology)? What are the limitations of the method? Were you able to detect them independently from the data coverage, dip angle and offset? Or there are preferential geometries that are emphasized from the data acquisition (a certain strike and deep for example)? Discuss all of this in the text.

This comment is not clear to us, which data are you talking about here? The legacy seismic data or the near surface VS model? The VS model is shown in many ways: in figure 10 all the line are shown in a 3D representation, some selected lines are reported in figure 12 to outline correlation with boreholes where available, zoom and relevant structures are shown in figure 11, while in figure 13 we show an interpolated velocity volume with several horizontal slices and identification of low velocity layers in the 3D volume. The lateral discontinuities are clearly evident in the horizontal slices. Finally, a smooth version of the velocity model is compared with deep legacy data and shows that dykes identified at mine level and only partly detected by the 3D seismic are confirmed in their shallow expression on the VS near surface model. We are open to suggestions about other ways to represent the data that may facilitate the identification of further features in addition to those that are highlighted. In terms of legacy data, we have decided to show more information on structures (not only dykes) imaged and relate that to those delineated in the Vs model.

Finally, you are using a new reflection active seismic data (3D and 2D), processing it for SW analysis and comparing with results from an older acquisition with different parameters not better specified, to show that you get information in the shallow area that you do not have in the 3D imaging. I agree with the statement but this comparison is not fair due to the different acquisition and of 20 years of

evolution in the seismic industry, instead, you should compare the results from the same acquisition and with the same parameters for a proper comparison. You can then suggest at which depth it is better the SW analysis and at which the seismic reflection imaging. And you should distinguish between the 2D and 3D cases, since they have different resolution and coverage, to distinguish in which cases this method is useful (maybe all of them but you need to show it). At the end, the main question is "Why should someone do SW analysis instead of the more traditional seismic reflection processing, since the acquired data is the same?", it is not faster nor cheaper, so you have to show that you get better results or extra information from the same acquisition.

We think the scope of the paper has not been fully understood so we have clarified in the introduction that we use the SW analysis not as a potential alternative to the seismic reflection data but as a complement to improve the knowledge of the near surface and derive information that are not available in the seismic reflection data per se. We do not understand why the legacy data, which are fully processed, should not be used to integrate the information retrieved by the near-surface model obtained. We understand the curiosity of the reviewer about the new 3D dataset and when it will be completely processed it is our intention to publish it. We also do not understand why the integrated interpretation between surface wave analysis and 3D seismic reflection could be done only with data acquired using the same acquisition parameters. The two methods use different signals, have different targets and different volume of interest, we therefore do not think that SW could not be used to provide additional information that is not observed on other available data. We have integrated the interpretation of these methods to overcome their pitfalls and leverage their strengths. In the conclusion of the paper we state that:

*"SW analysis, conducted on DSR surveys using MASW, was used to investigate the near-surface (top 360 m) at the vicinity of South Deep Gold Mine (South Africa) and investigate the effectiveness of integrating a near-surface S-wave velocity model with a deeper P-wave seismic reflection model, mine mapping and drilling information to aid in delineating possible water conduits."*
We think that this clarifies the aim of the present paper and we have strengthen this concept also in the introduction. Sentence 79 in the introduction has been edited to "This paper presents the applications, potentials and benefits of conducting SW analysis to reflection seismic data mainly acquired for deep targeting in a mining environment"

In addition, we have also provided some additional document that summarizes the previous research work done on the site and outstanding research questions.

Technical comments

We thank the reviewer for spotting typos and issues with references. We have addressed all these remarks. For those that instead requires a response, our reply is addressed here below point by point.

Line 68 – "Malehmir et al, 2015"

Done

Line 98 – Write "(Fig. 1) and not "(Fig. 1a and b)"

Done

Line 98-101 – Rewrite the phrase more clearly, consider to split it in 2 phrases. State clearly that the following phrases are a general setting and later state that you are going into detail of the interested area.

Done

Line 102 – It misses a "and" after the coma.

Done

Line 130 – Remove the symbol "~" from the age since you are already giving an error range. The use of the age unit in "Ga" for line 118 and "Ma" in this and the next line for the same order of magnitude is slightly confusing, I suggest to be consistent all along the text.

Done

Line 132 – "(Dorland, 2004)"

Done

Figure 2 – This figure needs some update. (1) a and b are too small, the text is not readable and the profiles are not distinguishable. (2) the legend in a is only for the geology while the seismic sensors are described in the caption, but then in b the legend is referred to the seismic sensors, and in none you are referring to the boreholes. You may try to combine a and b on a single figure with a single complete legend or remove the satellite view, it is interesting but not of high relevance.

This has been corrected

Line 167 – I think you mean "slip-sweep" and not "sleep-weep". Check all along the manuscript.

Done

Line 177 – You can define the distorted layout as crooked or moderately crooked profiles. Specify this especially for the 2D profiles.

Done

Table 1 – Fix slip-sweep and slip time. Only the sweep frequencies for the 3D grid are linear?

Done

Line 186 – Remove "(Fig. 2b; Table 1)". It is just confusing.

Done

Line 192 – "Clearbout and Green, 2008"

Done

Line 195 – "Clearbout and Green, 2008"

Done

Line 206 – "(Socco et al., 2009)" is in between two dots. Remove it or remove the first dot.

Done

Line 228 – "CR gathers" write it as previously in the text (CRGs).

Done

Line 220-230 – You state that the used spatial windows are mostly "relatively large", can you add what would have been the optimal or desired window size for the analyzed datasets?

We understand this statement is misleading and we modified it. What we meant was that, due to the receiver spacing, which is 25 m for receiver lines and 12.5 m for shot lines, the window is large with respect to usual near surface SW survey. Typically for NS investigations, data which are acquired on purpose for SW analysis would use receiver spacing of 2-5 m. In this case we use data acquired for a deep 3D seismic exploration and the survey design is not optimized for SW. This imposes to use processing windows larger than in traditional SW survey. We anyway did not mean to say that this is "not optimal" it is simply what is possible to do with the present data. Obviously, this is detrimental to lateral resolution as we stated at line 226.

Line 233 – With the provided figure no ground roll is "clearly visible". The picture is too small and the resolution low, maybe zoom the shot gather in the near offset if you do not need the far offset traces. In general, you do not use the term "clearly", it is not a very scientific term.

We think the lack of resolution of pictures was due to conversion to pdf in the previous submission. We have modified Figure 4, improved resolution, and zoomed image of the surface wavetrain has been added for improved clarity. The aim of the figure was to show a typical example of the seismic records obtained in the study area. The seismic records, however, characteristics of the study area are very varied, as shown in Figure 3. Likewise, the surface wave propagation and their clarity vary considerably, mainly depending on the geology. As seen in Figure 3 and 4 below, several dispersion curves that support the claimed investigations depth are obtained.

[Figure]

Figure 3: Typical seismic records obtained in the study area.

[Figure]

Figure 4: Typical dispersion curves obtained from the seismic record of the study area.

Line 234 – "after transforming the data" You may specify "the data in the red dashed rectangle" to do not confuse the reader.

Done

Figure 4 – The figure 4a is too small and low resolution, consider showing only part of it or adding a zoomed window. In figure 4d and e you use the Easting as x axes, but if I understood correctly they represent profile 1 and profile 2 that are perpendicular to each other and none of them parallel to the Easting direction. Did you consider to plot them versus the profile distance or one the easting and the other one the northing instead? It may be more representative of the real situation and easier to read in this case. Unless what you are showing is a projection of the profiles on EW direction, but in this case, you should explain it better in the text (same for figures 9a, 10 and 11).

A zoomed window has been added to Figure 4a. There is essentially no correct way to display a 2D section with arbitrary x and y coordinates using only one set of coordinates or distance as each point is uniquely identified by an ordered pair of coordinates. If you only use one set of coordinates or distance you are effectively mapping points from a 2D plane to a 1D line, losing the information about the second dimension. However, for analysis or visualization purposes, this is often done. In our case, we chose to project the data in an East-West orientation so as to preserve the geometric relation between the points, thus whether it is presented in the East-West or North-South orientation does not matter.

Line 249 – In figure 5 caption you may add "plotted as a function of wavelength and spatial location" as stated in the caption of figure 4 (remove depth), to keep consistent wording along the manuscript and make it easier for the reader to follow. Make the labels bigger in the figure.

That was a typo. Thanks for spotting it.

Line 272 – Add a reference or explanation for "(considering approximately depth = wavelength/2.5)."

We removed this sentence because we understood it was misleading with respect to the investigation depth of the inverted model

Figure 7 – Center "Phase velocity (m/s)" label.

Done

I suggest to consider the LCI chapter as a subchapter of data processing (4.1 instead than 5).

Done

Figure 8 – What is the inset shown in 8b? state it in the caption or show it in the figure (or both), I think it is a zoom of the Global misfit but it is not clear.

The labeling of the inset has been made clear and has been added to the caption, thank you.

Line 295 – In this study you used a laterally invariant initial model for the whole area. Since the area is quite big and heterogeneous, did you test how a more detailed initial model affects the results? and more in general, did you tested different initial models to discriminate dependencies of the results from the starting model itself? Discuss it

No, we did not test different initial models and we used a vertically smooth and laterally invariant initial model on purpose to avoid biases.

Line 311 – Fix "Figs. 9a and b)".

Done

Line 312 – What do you mean by "variations in the study area are resolved"? be more specific.

This has been clarified.

Line 315 – Fix "Figs. 9a and b)".

Done

Line 316 – Remove the dash and use parenthesis instead.

Done

Line 315-320 – Try to be clearer and more concise on what you want to highlight. What do you mean by "low- and high-velocity zones"? It is normal to have low and high velocity unless you are in a homogeneous media. Remove terms as "Evident" and "prominent" and use more scientific and quantitative terms (i.e., in figure 9c is visible a S-wave velocity reversal zone at around xx m of depths).

Done

Line 320 – Remove "(Fig. 9c)" if you already stated "In figure 9c" at the beginning of the phrase.

Done

Figure 9c – The 3D grid plot is very interesting with the full data view, but if you want to use it for highlighting the correlations between datasets it is not useful. The correlations you indicate in the circles are not visible and just take time for the reader to search for them. Or you modify the figure to make these areas more visible (remove some of the data, use zoom in windows, etc.) or you remove it from the figure and just state in the text and the reader will have to trust you. It is also very difficult to understand which data is from the 2D profiles and which one is from the 3D acquisition.

The figures have been modified to make these correlated areas more visible.

Figure 10 – In the legend write "inferred variations within the formations", using Hekpoort-Timeball Formation is confusing.

Changed to 'inferred geological variations within the formation'

Line 398 – How did you do the interpolation? Did you consider the different coverage and the different resolution of the datasets? Write it in the text.

This comment is unclear for us. The interpolation was carried out using a cubic interpolation technique.

Line 398-404 – Rephrase it to be clearer.

Done

Line 412-419 – Move to the "Site and data" chapter.

Done

Line 424 – If you "already stated" it there is no need to repeat. Give all needed information at the same point.

Done

Figure 13 – You do not state in the caption what MT stands for (I assume mine tunnels?).

Done

Figure 14 – You draw the dyke all the way to the surface, do you have geological evidences of it at the surface?

The dykes are not modelled all the way to the surface. They die below the base of the Transvaal Supergroup because the timing of their activity is unknown. The dykes are well understood at the mining level where they are exposed: they have been mapped, drilled, and sampled.
The low velocity zones are related to both faults and dykes, which are both possible causes of the observed near-surface low velocity zones as stated in the paper (see below)
"The low-velocity zones are possibly structurally weak zones that allowed for dyke emplacement (e.g., fracture zones or faults) or are the weathered tops of the dykes. The dykes are mafic in composition, which means that they are less competent in the near-surface compared to silicate rocks, and therefore break down more than the surrounding sedimentary rocks resulting in lower seismic velocities relative to the surrounding sedimentary rocks (Evans et al., 1998). Noted on the seismic sections is that the reflector disturbances associated with these faults and/or dykes are not visible. This could be because (1), in the case of faults, their throws are not large enough to be resolved by

the seismic data, and (2), in the case of the dykes, they are near-vertical with thickness less than the resolution limit of the legacy seismic reflection data"

Generally the dykes are in close proximity to fractures and fault zones, which correlate with the low-velocity zones. These fractures and fault zones acted as planes of weaknesses for preferential emplacement of the dykes. The dykes are most likely mafic in composition and are thus, less resistant to breakdown (weathering and alteration) compared to silicate rocks. In addition, the dykes that occur in the near-surface are more prone to weathering, weakening their original physical properties. A section on weathering in the study area has been included (see previous comment).

Line 518 – From what you showed and said so far, the S-wave velocity model does not show any dyke. It shows some low-velocity zone that you suggest may be related to dykes, but you need to provide more information as detailed in the general comments if you want to state this. The low-velocity zones could be just due to faults and local fracturing.

In fact, this is not entirely true. We give two explanations in the manuscript to explain the low velocity zone. We interpret the low velocity to be either associated with the fault (intruded by a dyke) or a weathered dyke. Both interpretations are scientifically possible, which is precisely what interpretation is all about. The Vs model provides us with the velocity, and we have to use the local geology knowledge to interpret the zone geologically.  In the discussion, we state that it is also possible that the low-velocity zone on the Vs model is due to faulting, rather than the dyke. However, the spatial correlation of this velocity zone and the dyke location (as confirmed through drilling and underground mapping) may suggest that the dyke might have intruded into the fault zone that crosscuts the overlying aquifers, implying that the deeper mining level and overlying aquifer systems are structurally connected, thus these structures may transport water to the mining level. This is one of the primary objectives of integrating legacy seismic data with SW analysis, namely, to enhance our understanding of the near-surface geology. We provide an alternative interpretation of the low-velocity zone that involves both faulting and dyke formation, so we don't understand why the reviewer is only focusing on the dyke interpretation, not the one related to faulting.

See the statement below from the paper.
"Noted on the seismic sections is that the reflector disturbances associated with these faults and/or dykes are not visible. This could be because (1), in the case of faults, their throws are not large enough to be resolved by the seismic data, and (2), in the case of the dykes, they are near-vertical with

thickness less than the resolution limit of the legacy seismic reflection data. These dykes and linked near-surface low-velocity zones are possible water source preferential flow-pathways that act as pathways for water migration from the Chuniespoort Group dolomites into the VCR and UER mining levels. Several similar dykes and faults that displace both the BLR and VCR in the area, but are not observed in the conventional seismic section, are reported in the works of Manzi et al. (2012)."

Line 538 – Compromised by what? Specify with some example.

Done

Line 539-541 – This statement is subjected to the replies and additional information you will provide respect to my specific comments.

We do not understand this remark.

Line 592 – I cannot find this reference in the text. Check and in case remove it.

It is in the introduction, line 71.

Line 604 – This reference is a copy of the one just above, remove it.

Done

Line 639 – I cannot find this reference in the text. Check and in case remove it.

Done

Line 657 – I cannot find this reference in the text. Check and in case remove it.

Done

---

## Author Comment (AC2)

Response to reviewer Milovan Urosevic

Dear Milovan Urosevic,

Thanks for the careful review of our manuscript. We have addressed all your remarks and below you find our point-to-point response outlined in blue.

Kind regards,

Sikelela Gomo                                                                    Johannesburg, 14/09/2025

(Corresponding Author)

RESPONSE

Specific comments

MASW: Claimed depth of investigation of 360 m seems to be justified with very few data points (Fig. 8). From most of the displays provided the maximum depth appears to be around 200-220m, which is more likely to be the case, based on the acquisition parameters and the spread length over which surface waves could be traced on the low-res images provided.

We think that the transformation of our manuscript in pdf for the previous submission generated too low resolution figures and this has created some reasonable doubts in the reviewers. We will be careful to provide higher resolution figures in the corrected manuscript. Looking at the obtained models (Figure 10, 11, 12, and 13 of the manuscript) it is apparent that there are several zones in the investigated area where the deeper layers in the model have been updated with respect to the initial model value, thus confirming that the data are sensitive to the model throughout the selected investigation depth. In Figure 7 of the manuscript, we report the spatial distribution of the datapoints at different wavelengths, showing that there is a significant number of dispersion curves with wavelength well beyond 360 m, which we assumed as bottom of our model. Also to respond to the doubts of Samuel Zappalà (RC1), we carried out a sensitivity analysis (reported here in Figure 1 and that we do not plan to add to the paper) starting from a monte Carlo inversion of a dispersion curve taken from the dataset. The sensitivity analysis shows that the dispersion curves with longer wavelength are sensitive to the subsurface properties at the depth of our models. We

also computed the wavelength/depth relationship according to Socco et al, 2017. This relationship depicts the skin depth of the surface wave propagation and confirms that the investigation depth is deeper than 200 m.

Nevertheless, we agree that only a limited portion of the curves reaches the wavelength needed to resolve the deeper layer of the model and that most of the estimated velocities below 300 m are generated by the spatially constrained inversion as a kind of interpolation of the available information at that depth. We have then decided to cut our model at 300 m. This change in our representation does not modify the interpretation that is based on the upper portion of the model.

[Figure]

Figure 1: Sensitivity analysis of the medium-frequency band dispersion curve from the dataset (Line P1) to velocity variations in the deeper layers. (a–b) Selected dispersion curves and corresponding VS models from 1D Monte Carlo inversion (Socco and Boiero, 2008). (c) W/D analysis (Socco et al., 2017) indicating the investigation depth as a function of wavelength. (d–f)

Sensitivity of the dispersion curve to a 30% perturbation of VS in the half-space. (g–i) Sensitivity to a 30% perturbation of VS in the layer above the half-space.

Socco L.V., C. Comina, F. Khosro Anjom, 2017, Time-average velocity estimation through surface-wave analysis: Part 1 — S-wave velocity: GEOPHYSICS, 82, 3, U49–U59, https://doi.org/10.1190/GEO2016-0367.1
Socco, L.V, D. Boiero, 2008. Improved Monte Carlo Inversion of Surface Wave Data: Geophysical Prospecting, 56, 357-371 https://doi.org/10.1111/j.1365-2478.2007.00678.x

Fig.8 phase velocities are reasonable. The inverted velocities go to 5000 m/s, seems much to high at depths of 200-300 m. Pretoria complex: Vp (5.0-6.5 km/s), Dolerite intrusions 6.0-6.8 Km/s. Hence Vs higher than 3.8 km/s can hardly be expected.

Thanks for this comment. We have thoroughly analyzed the results and in fact the value of 5000 m/s is associated to the deeper layer of the model which is the one that presents the lower sensitivity. We have hence changed our data representation and limit our representation to 4000 m/s. In Figure 2 we show some examples of lines with the new color scale and we show that this representation allows the investigation depth of the different portions of the model to be outlined. This change in the data presentation does not change the velocity model in the upper portion of the model on which our interpretation is based.

[Figure]

Figure 2. Vs profiles extracted from the 3D grid. The profiles are limited to shear Vs of 4000 m/s and illustrate the variable nature of the depth penetration of the data.

Receiver spacing of 10m risks spatial aliasing of surface waves. A frequency–wavenumber (F-K) analysis of both the newly acquired and legacy datasets would significantly strengthen the validation of the applied methods. If remedial steps were taken to mitigate aliasing risks, they should be documented to increase the confidence in the findings.

Rayleigh waves are highly energetic and easy to recognize in the spectra. If the source is in end off mode and hence the Rayleigh waves travel in one direction along the receiver spread used for the dispersion curve computation, this allows to compute the spectrum beyond the Nyquist

wavenumber. The aliased part of the propagating wavefield will then appear unwrapped and picking beyond Nyquist limitation will be possible (see Socco and Strobbia, 2004 for details). Anyway, in the present dataset, the information content in the high frequency band is very limited and in most of the picked dispersion curves we do not have energy in the wavelength range that would produce spatial aliasing. Here below we show a fk spectrum of a selected shot as example (Figure 3). The spectrum is computed up to 3 times the Nyquist wavenumber and we show that, with our spatial sampling, no aliasing is expected. To explain that the aliased part, if present, can be easily recovered, we resampled the same record with coarser spatial sampling and computed again the fk spectrum. The dispersion curve picked from the original data is superimposed on the spectrum and extends beyond the Nyquist wavenumber. We do not think to add this figure in the revised version of the paper, but we will add a sentence to explain that aliasing is not an issue even though the receiver spacing is quite large with respect to usual SW data.

[Figure]

Figure 3. f–k analysis for dispersion-curve picking along line P1 using a 300 m spatial window with (a) every receiver and (b) every 4th receiver. The f–k spectrum is unwrapped to extend the analysis beyond the Nyquist wavenumber. In (a), the black curve marks the picked dispersion curve, which is superimposed in (b) for comparison. (c–d) Dispersion curves expressed as functions of frequency and wavelength, respectively.

Socco, L.V., C. Strobbia, 2004, Surface-wave method for near-surface characterization: a tutorial: Near Surface Geophysics 2,4 165-185 https://doi.org/10.3997/1873-0604.2004015.

It remains unclear why P-wave refraction arrivals, which are visibly present in the provided shot records, were not incorporated into the geophysical analysis. Integrating these data with shear-wave profiles would offer a more robust inversion framework. Furthermore, by linking P- and S-wave velocity models, the authors had the opportunity to compute additional elastic parameters, particularly Poisson's ratio, which could enhance aquifer characterization.

We definitely agree that it will be interesting to compare P-wave data with the VS data from SW analysis. This comparison, with the addition of surface wave attributes (Colombero et al., 2019) that correlate very well with the velocity models, is the object of a paper in preparation. We decided not to include this part of the work in the present paper, as the actual paper is already quite extensive and lengthy. We also like the suggestion of computing the Posson's ratio from the computed P and S wave velocity models, and this will be incorporated in the future work. Future work also includes the SW analysis of the surface DAS and broadband MEMS-based sensors that were co-located with the data from the 5 Hz geophones presented here.

Colombero, C., C. Comina, L.V. Socco, 2019, Imaging near-surface sharp lateral variations with surface-wave methods — Part 1: Detection and location: Geophysics, 84, 6, EN93-EN111 DOI: https://doi.org/10.1190/geo2019-0149.1

Figure 4 shows Rayleigh waves that are barely visible and limited to a spread length of approximately 200 m. Based on standard MASW practices, this propagation range does not support the claimed investigation depth of 360 m, especially when employing 5 Hz geophones and assuming realistic shear wave velocities for the site.

We again are sorry for the low resolution of the first submission. We modified Figure 4 of the manuscript, improved resolution, and zoomed image of the surface wave train has been added for improved clarity. The aim of the figure was to show a typical example of the seismic records obtained in the study area. The seismic records, however, characteristics of the study area are very varied, as shown in Figure 4. Likewise, the surface wave propagation and their clarity vary considerably, mainly depending on the geology. As seen in Figure 4 and 5 below, several dispersion curves that support the claimed investigations depth are obtained.

[Figure]

Figure 4: Typical seismic records obtained in the study area.

[Figure]

Figure 5: Typical dispersion curves obtained from the seismic record of the study area.

Moreover, the vibroseis source utilized in the survey is known to produce relatively weak surface wave energy. Its long-duration sweep increases the risk of mixed wavefields, which may affect dispersion curve extraction.

We humbly disagree with this statement. There is no reason to think that vibroseis would produce a weak energy surface wavetrain. Our personal experience in comparing impact and vibrating sources did not evidence significant difference in the two kinds of source at the same site. We also made extensive test with a light vibrator with different sweep lengths and also analyzed the difference of the results obtained on raw or deconvolved data. No evident differences were found and longer sweeps always increased the data quality. Unfortunately, these analyses have never been used for a publication so we cannot provide a reference, but in Figure 6 we show two records that are part of those analyses. The first is the record of a 90 kg vibroseis shot and the second is the stack of 11 shot with a hammer. These data are not completely comparable to those acquired for the present study, but show that in the same condition vibrating sources generate data completely comparable with impact sources. It is also worth noting that the behavior of the source (coupling, harmonics, etc.) and the wavefield propagation are dependent on-site conditions.

[Figure]

Figure 6. At the top is a deconvolved record acquired with a light vibrating source, and at the bottom is a record obtained by stacking 11 hammer (5kg) source repetitions at the same source

point. These data are not related to the present work and are just aimed at supporting the above statement with experimental evidence.

Figure 6 exhibits well-developed P-wave refractions extending across the full 2 km spread. Surface waves appear to be present, at best over 200m length, and/or spatially aliased. The small display scale and low resolution of the field data images severely hinder a proper assessment of wavefield characteristics in both Figures 4 and 6.

Resolutions of figures in the manuscript have been improved. Due to the large number of Figures constituting the manuscript already, we avoided adding more images, however, examples of seismic records showing varied and large surface wave propagations in the analysed data are shown in Figure 4 above.

To allow for accurate interpretation of all wave types, significantly improved display quality is necessary. I recommend that representative shot records from both the new and legacy surveys be shown independently, without additional overlays, so that readers can assess signal quality and wavefield content with greater clarity.

We agree with this recommendation. All the raw data from both old and new datasets are available to the authors. However, we believe that this suggestion should be incorporated into the current work by the PhD candidate, which involves processing the latest 3D seismic data for comparison with the legacy 3D seismic data. The legacy data presented are included solely for final interpretation purposes, and SW analysis is provided to complement the legacy data and other geological data, characterizing the near-surface geology. Thus, the presentation of the shot records does not fall within the scope of the current paper and will definitely be considered for the future paper.

Figure 13 suggests that the legacy data may have undergone excessive (signal-to-noise ratio) SNR enhancement, potentially resulting in an overly smoothed image that masks subsurface discontinuities. If access to the raw legacy dataset is possible, reprocessing it with modern imaging

techniques could yield a significantly different structural interpretation. Moreover, the reprocessing may bring more clarity in the near surface, possibly up to 200 m depth.

This is possible and the legacy seismic is currently being re-processed using latest imaging techniques, particularly focusing on depth imaging. We also plan to analyze the diffractions on the stacked data that may be related to faults and dykes that are not well observed on the current legacy data. Additionally, we observe that the limitation for near-surface imaging may be related to the acquisition parameters (e.g., RI, RLI, SI, and SLI) and the bandwidth-limited sensors and sources used during data acquisition.

Lows in Vs at the position of interpreted dykes could be related to the computations since dolerite dykes are likely to have a higher velocity than the surrounding rocks.

In general, yes, but at South Deep Mine and the Witwatersrand goldfields, the dykes can be weathered and competent – these can vary along the length of the dyke. The dykes can be less competent in the near-surface compared to silicate rocks, and therefore breakdown more than the surrounding metasedimentary rocks, when exposed to weathering processes, resulting in lower seismic velocities relative to the surrounding metasedimentary rocks (Evans et al., 1998). The emplacement of dykes at South Deep mine often occurs along weak zones (i.e., faults and fractures). In these zones, the subsurface is often characterized by intensified weathering and alteration, due to the presence of increased fracturing, porosity, and fluids, which accelerate the breakdown of rock in these zones, including both in-situ rock and the associated dykes. Thus, it is possible that the dykes in the area are highly weathered. It is also possible that the low-velocity zone on the Vs model is due to faulting, rather than the dyke. Two alternative explanations are presented in the discussion section. However, the spatial correlation of this velocity zone and the dyke location (as confirmed through drilling and underground mapping) may suggest that the dyke might have intruded into the fault zone that crosscuts the overlying aquifers, implying that the deeper mining level and overlying aquifer systems are structurally connected, thus these structures may transport water to the mining level. This is the main objective of integrating the legacy and SW analysis, i.e., to enhance our understanding of the near-surface geology.

**Recommendations**

The investigative approach is conceptually appealing yet suffers from poor presentation of field data. High-quality visual displays are crucial for transparent interpretation. This needs to be corrected.

The quality of the displays presented has been improved.

I believe that the analysis would be strengthened by incorporating P-wave refraction tomography, given that refraction images already span the full spread length.
F–K (frequency–wavenumber) plots should be included to assess the integrity of surface wave sampling and verify spatial aliasing.

Please, see the responses to your previous remarks on this concern. We do not plan to add the fk that we have included in this response to the paper, since the number of figures is already very high and the paper very long. Moreover, we did not use fk for extracting dispersion curves so this would risk creating confusion about the dispersion images used in the processing. The improved resolution of the figures with respect to the previous submission should allow the reader to assess the presence of the surface waves in the raw data and on the dispersion images.
Concerning the analysis of P-wave travel times, beside the response above, it is useful to remark that the availability of P-wave arrival along the records does not guarantee a larger investigation depth with respect to that obtained by dispersion curve analysis and that low velocity layers that are retrieved from SW analysis in large portion of the area would be very difficult targets for P-wave analysis.

The stated depth of investigation needs more rigorous justification. I recommend including a brief overview of MASW methodology and the commonly accepted rule of thumb for depth estimation.

After working for more than 10 years on rigorous estimation of investigation depth of surface waves we tend to avoid using rule of thumbs but to assess directly the investigation depth from the data. The wavelength-depth plot presented on the response above (Figure 1) and the sensitivity analysis represent the skin depth of the data and show the capability of the chosen parameterization

to exploit the sensitivity of the data also at depth. We do not plan to report the sensitivity analysis in the paper but we will add a paragraph to discuss these aspects. Moreover, we have cut the model at 300 m.

The geology chapter could be much more concise and related to the analysis shown. Geological referencing is way to extensive for a little benefit to the reader.

The chapter has been revised to provide a summary of previous work done in the area, particularly related to the lack of near-surface characterization, and to focus on the timing of activity of faults and dykes in the region that link the shallow geology and deeper mining levels.

---

## Author Comment (AC3)

Dear Giacomo Medici

Thanks for the careful review of our manuscript. Here below we have responded to all your remarks.

Please find below our comments outlined in blue

Kind regards

Sikelela Gomo                                                                 Johannesburg, 14/07/2025

(Corresponding Author)

RESPONSE

Line 37. You use multiple times "conduits". This word has a specific meaning in karst hydrology. Thus, I suggest an universal change into "perferential flow-pathways"

Done

Lines 60-63. "Near-surface characterization…environmental, and civil engineering infrastructure". Insert recent literature on recent near-surafce hydrogeophysical charaterization in fractured/faulted bedrocks.

- Medici, G., Munn, J.D., Parker, B.L. 2024. Delineating aquitard characteristics within a Silurian dolostone aquifer using high-density hydraulic head and fracture datasets. Hydrogeology Journal, 32, 1663-1691.

- Svetina, J., Prestor, J., Mozetič, S., & Brenčič, M. (2025). Ambient intraborehole flow in a highly productive aquifer in Ljubljana, Slovenia. Journal of Hydrology: Regional Studies, 57, 102139.

We thank for the interesting suggestions, but after reading the papers we do not think they are relevant for the present case since they do not refer to the investigated region and formations. Moreover, reviewers invited by the editor have commented that the geological and general description is already lengthy and has possibly redundant references.

Lines 92-94. You have clarified the 3 specific objectives. What about the general goal?

We agree that the general goal of the paper was not properly expressed. We have revised the abstract, the introduction and discussion and conclusion to clarify the multiple methodological and specific objectives addressed by the paper. We have hence clarified that the paper has a methodological objective that is maximizing the information that can be extracted from the groundroll in seismic reflection data to provide a VS model of the near surface to a significant depth. To increase the illumination we used the data both in common shot gather and in common receiver gather configuration (which is on our knowledge new in the field of surface wave data analysis) and we pushed the processing to the limit of resolution and investigation depth. The availability of a VS model to a depth of about 300 m allowed us to carry out a structural and geological interpretation connecting the information from deep exploration and in mine evidences with the near surface structure and providing the prove that dikes known to crosscut the formation at mine depth reach the surface. These aspects have been better explained.

Line 135. "Slump faults". Please, clarify the nature of these faults. What about the presence of extensional, strike-slip and thrusts?

The paragraph containing sentence 135 has been edited to reflect/mention the structures enquired about. Generally, the Transvaal Supergroup shows evidence of both extensional and thrust faulting (Cousins, 1962).

Lines 135-145. Please, provide more detail on the type and genesis of the tectonic structures given the nature of the manuscript.

To address this point, the above-mentioned section of the manuscript has been modified to the following:

"The Transvaal Supergroup is displaced by post-Transvaal Supergroup age faults (extensional faults) associated with the Vredefort Impact Crater (Cousins, 1962) and is intruded by massive suites of mafic and ultramafic rocks (i.e., dyke swarms and sill provinces) thought to be 'pre'-'syn'- and 'post'-Bushveld Complex in age (Willemse, 1959; Sharpe, 1982, 1984; Schreiber et al., 1992; Gumsley et al., 2017). Additionally, the rocks of the Pretoria Group are displaced by post-Pretoria Group faults (i.e., both extensional and thrust faults) that displace both the Pretoria Group rocks and the base of the Transvaal Supergroup, and slump-faulting (extensional faults), which only affects the formations within the Transvaal Supergroup (Cousins, 1962). The slump faulting does not transgress beyond the base of the Transvaal Supergroup and is thought to originate from the subsidence of the Chuniespoort Group of the Transvaal Supergroup. Thus, the tectonic structures in this study are the post-Transvaal Supergroup age faults (extensional faults) followed by the post-Pretoria Group age faults (extensional and thrust faults). The post-Pretoria Group extensional faults include the slump-faulting. The post-Transvaal Supergroup/Pretoria Group age faults and intrusions are prone to mining-induced seismicity and influence the hydrology of the West Rand acting as pathways for water migration from the overlying Chuniespoort Group dolomitic units down to the gold reefs' mining levels (~3.5 km below ground surface) (Van Niekerk and Van Der Walt, 2006; Manzi et al., 2012). The intersection of these water conduits during mining often negatively affects the productivity of the mine and increases the safety risk to mine personnel and infrastructure. Thus, their delineation is important in ensuring mine safety, longevity, and increased productivity."

Lines 155- 279. Paragraphs 3, 4 and 5 look three sub-paragraphs of a "methodology". I suggest 3.1, 3.2, and 3.3.

Done

Line 412. "Investigate the structural linkage". Do you mean "structural geology linkage"? Please, specify.

Yes, this is what is meant and has been corrected.

Figures and tables

Figure 1a. The type of the northern fault is clear (strike-slip), not for the other tectonic lines. Can you improve main body, figure and caption on this point?

We looked at this thoroughly, however, the rest of the lineaments shown on the map most likely represent fault zones. Therefore, they are not associated with a single/simple fault line, but a broader zone of deformation made up of parallel or subparallel multi-fault segments related to a rich tectonic history (e.g., Colesberg lineament and the Thabazimbi-Murchison lineaments), thus we cannot associate them with one orientation.

Figure 1b. Labels and words that describe the stratigraphic column are not readable.

These have been improved

Figure 2. Insert the age of the lithotypes in the legend.

The legend in Figure 2 has been modified to show the age of the present formations.

Figure 14. Same issue here, labels and words that describe the geological model are not readable.

Corrected

---

## Author Comment (AC4)

**Some general comments to the reviewers are provided, offering context for the geology and seismic interpretation, as well as motivation for the use of legacy seismic data in this study.**

The 2003 D Kloof-South Deep (covering Kloof and South Deep Mine) data (Campbell and Crotty, 1990, Manzi et al., 2012a,b) have been the subject of several studies over the past 20 years, given the size of the survey area and the unique nature of the data, particularly the highly continuous seismic reflections associated with lithological contacts and the mapping of complex geological structures that cross the gold deposits. To mention a few that are relevant to this paper:

(1) Manzi (2012a,b, 2013a) reprocessed the data and computed seismic attributes to (a) delineate geological structures (faults and dykes) that cross-cut the Ventersdorp Contact Reef (gold deposit) and have the potential to migrate water and methane gas from the Transvaal Supergroup aquifers to the mining levels (~ 3 km). In Manzi et al. (2012b), the base of the 2.65–2.05 Ga Neoarchean–Paleoproterozoic Supergroup (Transvaal Supergroup) horizon (Black Reef, ~ 500 m below ground surface) and the mining horizon (VCR, ~ 3000 m below ground surface) were well imaged. The geological structures (faults and dykes) that were seismically defined at both Black Reef and VCR levels were spatially correlated with water and methane data obtained from drilling data at the mining levels. The integrated data showed a good spatial correlation (see Manzi et al., 2012b). The authors hypothesized that the water migrated from the overlying acquirers above the Black Reef (of the Transvaal Supergroup) and travelled through faults and dykes to the mining levels (within Witwatersrand Supergroup). However, due to the low resolution of the legacy seismic data in the near-surface (Manzi et al., 2012a, b; 2013a), the continuity of these structures (including the South Deep faults and dykes (especially those intruded in the fault zones)) was never investigated (see Figure 7 below of Manzi et al., 2013b). The manuscript presented here by Gomo et al. (in this special issue) attempts to provide some seismic constraints on the near-surface geology and previous interpretations: (1) the near-surface as derived from the Vs models exhibits structurally complex geology interpreted to be associated with shales, quartzites, basalts, sills, faults, and dykes. The interpretation is constrained by the geological information (surface mapping and boreholes), and (2) the Vs models, for the first time, provide some evidence that underlying structures (underground mapped and confirmed through drilling) crosscut the base of the Transvaal, meaning that they

crosscut the known groundwater aquifers within the base of the Transvaal Supergroup. The Vs model suggests that the mining level is structurally connected to the near-surface aquifers, making these geological features potential conduits for water migration (Manzi et al. 2012b).

(2) Following the work by Manzi et al. (2012a,b; 2013a) on the imaging of geological structures at the Black Reef and Mining levels, Manzi et al. (2013b) merged the re-processed 2003 reflection seismic data with other historical 3D seismic datasets (covering other mines in the Witwatersrand gold fields) to produce one continuous cube. The merged data were integrated with the geochronology data, seismic data, underground mapping data, surface and underground borehole data from the mines to build a tectonic model of the Witwatersrand Basin. The final model was constrained to the pre-Transvaal Supergroup (see Figure 2 below from Manzi 2013b), where all the structures were interpreted as pre-dating the Transvaal Supergroup. This published model will need to be revisited in the future to incorporate the new information that integrates legacy, geological information, and surface wave analysis (presented in this manuscript).

[Figure]

Figure 7. North–northeast regional crossline seismic section (line AA′ in Figure 6) through WUDLs, Driefontein, Kloof, and South Deep surveys. The West Rand and Bank Faults do not breach the BLR. The merge boundary between WUDLs, Driefontein, and Kloof-South Deep surveys is identifiable in the seismic section.

Figure 7, taken from Manzi et al. (2013b), show the regional structural architecture of the Witwatersrand Basin. Faults in this model pre-dates the Transvaal Supergroup.

[Figure]

**Fig. 2.** Seismic model across the Witwatersrand goldfields. (a) 3D regional seismic model incorporating the BLR, Ventersdorp lavas, VCR and B. Shale. The model shows the geometry of West Rand Fault, Bank Fault and Libanon Anticline. (b) Regional crossline seismic section (line AA' in Fig. 1) through WUDLs, Driefontein, Kloof and South Deep surveys, showing WRF and BF zones and their adjacent TT and JT, respectively. WRF: West Rand Fault; BF: Bank Fault; TF: Tandeka Thrust; JT: Jabulani Thrust; WRG: West Rand Group; CRG: Central Rand Group; Klip: Klipriviersberg; VCR: Ventersdorp Contact Reef; BLR: Black Reef Formation; B. Shale: Booysens Shale.

Figure 2 illustrates the tectonic model of the Witwatersrand Basin and study area, as derived from legacy seismic data, geological information (including borehole and mapping data), and geochronological data. Major faults in this model pre-dates the Transvaal Supergroup.

(3) The work by Nwaila et al. (2020) looked at the potential to mine the Black Reef horizon at Driefontein mine (26 km west of the South Deep mine). As part of this work, the authors investigated the structural architecture of the Transvaal Supergroup (same units as those found in South Deep) using the merged legacy seismic data (Manzi et al., 2013b). The seismic data were interpreted in an integrated approach with petrography, 3D micro-X-ray computed tomography, and machine learning to characterise the ore resource and understand the structural styles within the Transvaal Supergroup. The study revealed that some geological structures affecting the Black Reef (i.e., the base of the Transvaal at a depth of 500 m) can be traced downward into the underlying, older auriferous horizons (e.g., VCR at a depth of 3.0 -4.0 km). The outcome of this research provided some clues indicating that the near-surface aquifers at South Deep require further investigation. As already mentioned, the quality of the seismic data at South

Deep was not adequate to characterise the near-surface groundwater aquifer system. The current data, in its current form, provides high-resolution imaging of geological structures from 400 m to 6,000 m depth below the ground surface. The approach of surface wave analysis presented in this paper is intended to supplement the legacy reflection seismic data, borehole data, and mine geological model, rather than compare or replace these methods – it will not be a good approach to compare them, as they are different datasets designed and implemented at the site for different scientific and mining purposes. Really, the question is simple – is there a connection between the near-surface aquifers and mining horizon levels at the South Deep mine study area? Based on the Vs model presented in this manuscript, in conjunction with the legacy data and other datasets, our interpretation is that the two systems are connected at the South Deep mine.

(4) At Kloof Gold Mine, a mine covered by the same 2003 legacy data and located about 5 km away from the South Deep mine, water and methane-gas bearing geological structures analysed through seismic data were found to connect the Black Reef level and mining VCR level (Manzi et al., 2012b). To investigate the source of water and methane gas at the mining levels along these structures, water and methane gas samples were collected along the structures for isotope analysis (Manzi et al, 2016). The isotope and geochemistry results for water indicated the mixing between meteoric/shallow aquifer water and deep hypersaline water. Similarly, there was mixing between biotic (methane from overlying aquifers) and abiotic methane (methane from the deeper crust). This provided additional evidence that the water-methane-bearing structures were transporting water/methane gas (from the surface/near-surface and the deeper part of the crust) to the mining level. Future studies at South Deep will explore similar approaches, with interpretation constraints applied to the shallow part, utilizing surface wave analysis.

(5) Tectonically, the Transvaal Supergroup (target for this study) is structurally complex and has undergone several episodes of deformation (Kinsman, 1975; Veevers, 1981; Vermaakt and Chunnet, 1994; Coward et al., 1995; Manzi et al., 2013a,b; Nwaila et al., 2022). The lithological composition and geological setting are identical to those in, and thus considered to be the result of, rift-type tectonic settings formed by stretching or thinning the continental lithosphere (Allen et al., 2015). Pre-Transvaal Supergroup, the

Witwatersrand Basin generally exhibits major listric normal faults and their related drag folds that are ascribed to the extensional tectonic Platberg Volcanism at 2754-2709 Ma (Manzi et al., 2013b, Gumsley et al., 2020). Our study area is also characterised by numerous sills and dykes that intruded at different geological ages and have been subjected to several investigations. The sills and dykes are of various ages, such as known post-Karoo dykes of pre-Cretaceous and Cretaceous age (145 - 66 Ma), Karoo dykes (150 Ma), Pilanesberg (1.30 Ga), Vredefort meteorite impact event (2.05 Ga), the Bushveld Igneous Complex magmatism (2.03 Ga), Transvaal (2.20 Ga), and Ventersdorp (2.60 Ga) (Frimmel, 2014; Fuchs et al., 2016; Frimmel & Nwaila, 2020). The post-Black Reef tectonic events likely led to the reactivation of pre-existing faults, which are clearly imaged in the legacy seismic data covering South Deep mine and other neighbouring mines (Manzi et al., 2013; Nwaila et al., 2020a). Movement along faults inevitably results in the creation of numerous microfractures, which can be observed and mapped underground (if exposed). Current mining activities can also reactivate faults and dykes (Masethe et al., 2023), promoting water infiltration, movement, and circulation, and these can also host large and damaging seismic events.

(6) In summary, at the mining level of the South Deep mine, the faults, dykes, and fractures (some observed on legacy seismic data) are well-known and constrained by the drilling data and underground mapping (where they have been exposed underground). The dykes that are seismically active are well defined by the large seismic events (by studying the strike, focal mechanisms of the seismic events). What is not known is the continuity of these structures to the shallow aquifers. Current drilling programs are focused on underground drilling; near-surface information is limited to a few historical surface boreholes. Furthermore, it is a common practice in the mining industry not to log the core in detail in the zones with no mineralisation. At South Deep, it is not surprising that the surface drill cores lack detailed information in the near-surface.  For example, it is not within the interest of the mine geologists to understand the degree of weathering and fracturing of rocks that do not host mineralisation or are too far (in depth, more than 500 m) from the horizon of interest. Generally, the focus will be on lithological contact, major faults, and thicknesses. Thus, surface wave analysis attempts to characterize the near-surface geological structures (which is not well understood) and link it with the underground mining horizon (which is better understood) for current and future mine planning, development, and safety.